# CONFLICT-AVERSE GRADIENT AGGREGATION FOR CONSTRAINED MULTI-OBJECTIVE REINFORCEMENT LEARNING

**Dohyeong Kim, Mineui Hong, Jeongho Park, and Songhwai Oh**[*]
Dep. of Electrical and Computer Engineering and ASRI, Seoul National University

## ABSTRACT

In real-world applications, a reinforcement learning (RL) agent should consider multiple objectives and adhere to safety guidelines. To address these considerations, we propose a constrained multi-objective RL algorithm named *Constrained Multi-Objective Gradient Aggregator (CoMOGA)*. In the field of multi-objective optimization, managing conflicts between the gradients of the multiple objectives is crucial to prevent policies from converging to local optima. It is also essential to efficiently handle safety constraints for stable training and constraint satisfaction. We address these challenges straightforwardly by treating the maximization of multiple objectives as a constrained optimization problem (COP), where the constraints are defined to improve the original objectives. Existing safety constraints are then integrated into the COP, and the policy is updated by solving the COP, which ensures the avoidance of gradient conflicts. Despite its simplicity, CoMOGA guarantees convergence to global optima in a tabular setting. Through various experiments, we have confirmed that preventing gradient conflicts is critical, and the proposed method achieves constraint satisfaction across all tasks.

## 1 INTRODUCTION

Real-world reinforcement learning (RL) applications involve multiple objectives and have to consider safety simultaneously. For instance, legged robot locomotion tasks focus on maximizing the success rate of reaching goals and energy efficiency while avoiding collisions, as illustrated in Figure 1. To effectively tackle these considerations, constrained multi-objective RL (CMORL) has been introduced (Huang et al., 2022). CMORL aims to find a set of *constrained-Pareto (CP)* optimal policies (Hayes et al., 2022), which are not dominated by any others while satisfying safety constraints, rather than finding a single optimal policy. This allows users to choose and utilize their preferred policy from the set without additional training. In order to express a set of CP optimal policies, it is common to use a concept called *preference* (Cai et al., 2023; Kyriakis & Deshmukh, 2022), which encodes the relative importance of the objectives. Then, a set of policies can be represented by a preference-conditioned policy, a mapping function from a preference space to a policy space.

In most multi-objective RL (MORL) algorithms (Désidéri, 2012; Yang et al., 2019; He et al., 2024), the preference-conditioned policy is trained by maximizing a scalarized reward, computed as the dot product of a given preference and rewards of the multiple objectives. These approaches, called *linear scalarization (LS)*, offer the benefits of straightforward implementation and ensure Pareto-optimal convergence in tabular settings (Lu et al., 2023). However, when applied to deep RL with function approximators, LS tends to converge to local optima due to the nonlinearity of the objective functions, as shown in a toy example in Figure 2. In the field of multi-task learning (MTL) (Yu et al., 2020; Liu et al., 2021; Navon et al., 2022), this issue has been addressed by avoiding conflicts between the gradients of multiple objective functions. Building on these advancements, we extend these gradient-conflict avoiding techniques to CMORL, anticipating analogous benefits.

It is also crucial to handle constraints as well as multiple objectives in CMORL. A straightforward approach is treating the constraints as other objectives and concurrently adjusting the preferences corresponding to those objectives to maintain them below thresholds, as done in (Huang et al.,

---

[*]Corresponding author: songhwai@snu.ac.kr.

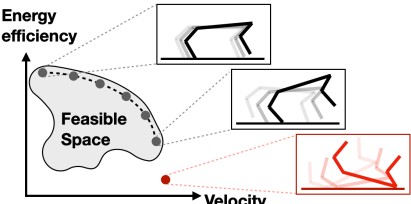

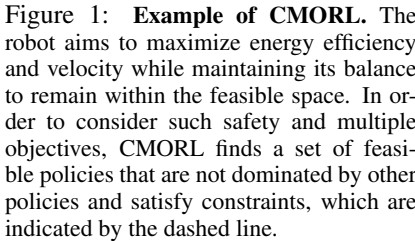

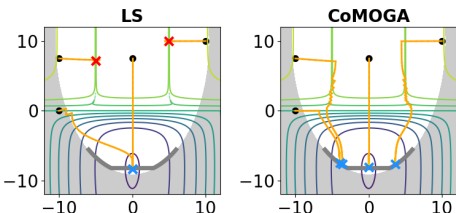

Figure 1: **Example of CMORL.** The robot aims to maximize energy efficiency and velocity while maintaining its balance to remain within the feasible space. In order to consider such safety and multiple objectives, CMORL finds a set of feasible policies that are not dominated by other policies and satisfy constraints, which are indicated by the dashed line.

Figure 2: **Comparison between linear scalarization (LS) and the proposed method (CoMOGA).** Optimization trajectories are shown in orange, with initial points marked by black circles and 'x' markers for the reached points. Contours and shaded areas represent average values of the two objective functions and regions where constraints are violated, respectively. The grey line indicates the CP optimal set. Unlike LS, CoMOGA consistently reaches the optimal set from any starting position. For more details, see Appendix A.

2022). This approach exactly aligns with the Lagrangian method, which solves the Lagrange dual problem by updating policy and multipliers alternatively. However, the Lagrangian method can make training unstable due to the concurrent update of policy and multipliers (Stooke et al., 2020; Kim & Oh, 2022). As the updates of the policy and multipliers influence each other, any misstep in either can make the training process diverge. Thus, it is critical to manage constraints without introducing additional optimization variables.

In order to avoid gradient conflicts and efficiently handle safety constraints, we introduce a novel but straightforward algorithm named ***Constrained Multi-Objective Gradient Aggregator (CoMOGA)***. CoMOGA treats a multi-objective maximization problem as a constrained optimization problem (COP), where constraints are designed to enhance the original objective functions in proportion to their respective preference values. Existing safety constraints are then incorporated into the constraint set of the COP. By explicitly preventing the given objectives from decreasing, CoMOGA successfully avoids gradient conflicts. The policy gradient is then computed by solving this reformulated problem within a local region, as in TRPO (Schulman et al., 2015), thereby eliminating the need for extra variables to handle safety constraints. While the proposed method is intuitive and straightforward, we have also shown that it converges to a CP optimal policy in a tabular setting.

The proposed method has been evaluated across diverse environments, including multi-objective tasks with and without constraints. The experimental results confirmed that avoiding gradient conflicts is effective in preventing convergence to local optima. This is particularly essential in environments with unstable dynamics, such as bipedal and humanoid locomotion tasks, since they are prone to get stuck in suboptimal behaviors, such as frequent falling. Furthermore, the absence of additional variables facilitated stable training while ensuring constraint satisfaction across all tasks.

## 2 RELATED WORK

**Multi-Objective RL.** MORL aims to find a set of Pareto-optimal policies that are not dominated by any others. Given that Pareto-optimal policies can be obtained with appropriate reward scalarization (Hayes et al., 2022), many MORL algorithms utilize scalarization techniques. Based on linear scalarization, Yang et al. (2019) introduced a new Bellman operator for MORL algorithms, which estimates a target Q value for a given preference by combining Q values from other preferences. Basaklar et al. (2023) presented a similar Bellman operator and extended it to an actor-critic framework to enable its application in continuous action spaces. For the continuous action spaces, Lu et al. (2023) proposed a SAC-based MORL algorithm, theoretically demonstrating that elements of the Pareto-optimal policy set can be achieved through linear scalarization. Not only the linear scalarization, Van Moffaert et al. (2013) proposed a nonlinear scalarization method, called *Chebyshev* scalarization, which can cover a non-convex policy set. To further improve performance, the evolution strategy (ES) can be applied as done in multi-objective optimization criteria (Deb et al., 2000). Chen et al. (2020) and Xu et al. (2020) proposed methods that repetitively update a policy for each population using linear scalarization and produce a new generation of policy parameters.

There is also a different approach from scalarization. Abdolmaleki et al. (2020) have proposed a heuristic method of applying preference in the action distribution space instead of the reward space. It is further extended to a CMORL algorithm in (Huang et al., 2022).

**Constrained RL.** Constrained RL is developed to explicitly consider safety or constraints in RL. Based on the approach to handling constraints, constrained RL algorithms can be categorized into two types: primal-dual methods and primal methods. The primal-dual methods, also called Lagrangian methods, are designed to address Lagrange dual problems. Ding et al. (2022) introduced a natural policy gradient-based primal-dual method and demonstrated its convergence to an optimal policy at a specified convergence rate. Another primal-dual method, proposed by Bai et al. (2022), ensures that a trained policy results in zero constraint violations during evaluation. While these methods are straightforward to implement and can be integrated with standard RL algorithms, the training process can be unstable due to additional optimization variables (Lagrange multipliers) (Stooke et al., 2020). In contrast to the primal-dual method, the primal method addresses the constrained RL problem directly, so no additional optimization variables are required. Achiam et al. (2017) introduced a trust region-based primal method. This approach linearly approximates a constraint within a trust region and updates the policy through a line search. Xu et al. (2021) presented a natural policy gradient-based algorithm and demonstrated its convergence rate towards an optimal policy. In multi-constrained settings, Kim et al. (2023) proposed a primal method to handle infeasible starting cases.

## 3 BACKGROUND

**Constrained Multi-Objective MDP.** We define a constrained multi-objective Markov decision process (CMOMDP) as a tuple represented by $\langle S, A, \mathcal{P}, R, C, \rho, \gamma \rangle$, where $S$ is a state space, $A$ is an action space, $\mathcal{P} : S \times A \times S \mapsto [0, 1]$ is a transition model, $R : S \times A \times S \mapsto \mathbb{R}^N$ is a vectorized reward function, $C : S \times A \times S \mapsto \mathbb{R}^M$ is a vectorized cost function, $\rho$ is an initial state distribution, and $\gamma$ is a discount factor. The reward and cost functions are bounded by $R_{\max}$. In the CMOMDP setting, a policy $\pi(\cdot|s) \in \Pi$ can be defined as a state-conditional action distribution. The objective function of the $i$th reward is defined as:

$$J_{R_i}(\pi) := \mathbb{E}\left[\sum\nolimits_{t=0}^{\infty} \gamma^t R_i(s_t, a_t, s_{t+1})\right],$$

where $s_0 \sim \rho$, $a_t \sim \pi(\cdot|s_t)$, and $s_{t+1} \sim \mathcal{P}(\cdot|s_t, a_t) \ \forall t$. We also define the constraint function $J_{C_k}(\pi)$ by replacing the reward with the cost function. A constrained multi-objective RL (CMORL) problem is defined as:

$$\text{maximize}_\pi \ J_{R_i}(\pi) \ \forall i \in \{1, ..., N\} \quad \textbf{s.t.} \ J_{C_k}(\pi) \leq d_k \ \forall k \in \{1, ..., M\}, \tag{1}$$

where $d_k$ is the threshold of the $k$th constraint. Here, it is ambiguous to determine which policy is optimal when there are multiple objectives. To address this, we instead define a set of optimal policies using the following notion, called *constrained dominance* (Miettinen, 1999):

**Definition 3.1** (Constrained Dominance). *Given two policies* $\pi_1, \pi_2 \in \{\pi \in \Pi | J_{C_k}(\pi) \leq d_k \ \forall k\}$, $\pi_1$ *is dominated by* $\pi_2$ *if* $J_{R_i}(\pi_1) \leq J_{R_i}(\pi_2) \ \forall i \in \{1, ..., N\}$.

A policy $\pi$ which is not dominated by any policy is called a *constrained-Pareto (CP) optimal* policy, and the set of all CP optimal policies is called *constrained-Pareto (CP) front* (Miettinen, 1999). Additionally, a condition for gradient conflicts is defined as follows:

**Definition 3.2** (Gradient Conflict). *Let* $g_i$ *be the gradient of the* $i$th *objective function, and let* $g$ *denote the policy update direction. We say there is a gradient conflict if* $\exists i$ *such that* $g_i^T g < 0$.

As studied in MTL (Liu et al., 2021; Yu et al., 2020) and shown in Figure 2, gradient conflicts can cause convergence to local optima. Thus, we aim to find the CP front while updating policies without gradient conflicts.

**Preference.** In order to express a subset of the CP front, prior MORL works (Xu et al., 2020; Alegre et al., 2023) introduce a preference space, defined as: $\Omega := \{\omega \in \mathbb{R}^N | \ \omega_i \geq 0, \ ||\omega||_1 = 1\}$, where a preference $\omega \in \Omega$ serves to specify the relative significance of each objective. Subsequently, a specific subset of the Pareto front can be characterized by a function mapping from the preference space to the policy space. We denote this function as a *universal policy* $\pi(\cdot|s, \omega)$, characterized by an action distribution that depends on both the state $s$ and preference $\omega$ variables. Ultimately, our goal is to train the universal policy that covers the CP front as extensively as possible.

# 4 PROPOSED METHOD

We propose constructing a sub-problem designed to aggregate gradients without conflicts. Building sub-problems is a widely used strategy in conflict-averse MTL approaches (Navon et al., 2022; Yu et al., 2020; Liu et al., 2021), but applying this strategy in the context of CMORL requires considering preferences of multiple objectives and ensuring the satisfaction of constraints. This section outlines the proposed method in four distinct parts: **1)** building a constrained optimization problem to handle the multiple objectives with safety constraints, **2)** calculating a policy gradient by aggregating gradients of the objective and constraint functions for a single preference, **3)** introducing a method for training a universal policy covering a spectrum of preferences, and **4)** concluding with an analysis of the convergence properties of the proposed method. Before that, we first parameterize the policy and define gradients of the objective and constraint functions, as well as a local region in the parameter space. By representing the policy as $\pi_\theta$ with a parameter $\theta \in \Theta$, the gradients of the objective and constraint functions in (1) are expressed as:

$$g_i := \nabla_\theta J_{R_i}(\pi_\theta), \ b_k := \nabla_\theta J_{C_k}(\pi_\theta).$$

For brevity, we will denote the objectives as $J_{R_i}(\theta)$ and the constraints as $J_{C_k}(\theta)$. The local region is then defined using a positive definite matrix $H$ as: $||\Delta\theta||_H \leq \epsilon$, where $||x||_H := \sqrt{x^T H x}$, and $\epsilon$ is the local region size, a hyperparameter. $H$ can be the identity matrix or the Fisher information matrix, as in TRPO (Schulman et al., 2015).

## 4.1 REFORMULATION AS CONSTRAINED OPTIMIZATION

We introduce a process of treating the CMORL problem as a constrained optimization problem (COP). To provide insight, we first present the process in a simplified problem where only the $i$th objective exists and then extend it to the original problem. The problem of maximizing the $i$th objective within the local region is expressed as follows:

$$\max_{\Delta\theta} J_{R_i}(\theta_{\text{old}} + \Delta\theta) \quad \text{s.t. } ||\Delta\theta||_H \leq \epsilon. \tag{2}$$

Assuming that $J_{R_i}(\theta)$ is linear with respect to $\theta$ within the local region, the above problem has the same solution as the following problem:

$$\min_{\Delta\theta} \Delta\theta^T H \Delta\theta \quad \text{s.t. } J_{R_i}(\theta_{\text{old}} + \Delta\theta) - J_{R_i}(\theta_{\text{old}}) \geq \epsilon ||g_i||_{H^{-1}} =: e_i. \tag{3}$$

It is shown that the solutions to (2) and (3) are identical in Appendix B.1, and this finding confirms that objectives can be converted into constraints in this way. Consequently, the CMORL problem can be reformulated as follows:

$$\min_{\Delta\theta} \Delta\theta^T H \Delta\theta \quad \text{s.t. } \omega_i e_i \leq J_{R_i}(\theta_{\text{old}} + \Delta\theta) - J_{R_i}(\theta_{\text{old}}) \ \forall i, \ J_{C_k}(\theta_{\text{old}} + \Delta\theta) \leq d_k \ \forall k, \tag{4}$$

where $e_i$ in (3) is scaled to $\omega_i e_i$ to reflect the given preference $\omega$. We will show that the policy updated using (4) converges to a CP optimal policy of (1) in Section 4.4 under mild assumptions.

## 4.2 GRADIENT AGGREGATION

Now, we compute a policy gradient by solving (4) within the local region using a linear approximation, similar to TRPO (Schulman et al., 2015). First, the problem (4) can be linearly approximated as the following quadratic programming (QP) problem:

$$\bar{g}_\omega^{\text{ag}} := \text{argmin}_{\Delta\theta} \Delta\theta^T H \Delta\theta \quad \text{s.t. } \omega_i e_i \leq g_i^T \Delta\theta \ \forall i, \ b_k^T \Delta\theta + J_{C_k}(\theta_{\text{old}}) \leq d_k \ \forall k, \tag{5}$$

where the gradients of the objective and constraint functions, $g_i$ and $b_k$, are aggregated into $\bar{g}_\omega^{\text{ag}}$. The gradient is then clipped to ensure that the policy is updated within the local region, as follows:

$$g_\omega^{\text{ag}} := \min(1, \epsilon/||\bar{g}_\omega^{\text{ag}}||_H)\bar{g}_\omega^{\text{ag}},$$

where the policy will be updated by $\theta_{\text{old}} + g_\omega^{\text{ag}}$. Note that the aggregated gradient satisfies $0 \leq g_i^T g_\omega^{\text{ag}}$ $\forall i$ due to the reformulated constraints in (5). This ensures that *the aggregated gradient does not conflict with any of the objective functions*. However, when both the constraints and objectives are reflected in the updates, the updated policy can violate the constraints by overly focusing on objective improvements. To address this issue, we take a recovery step which updates the policy by

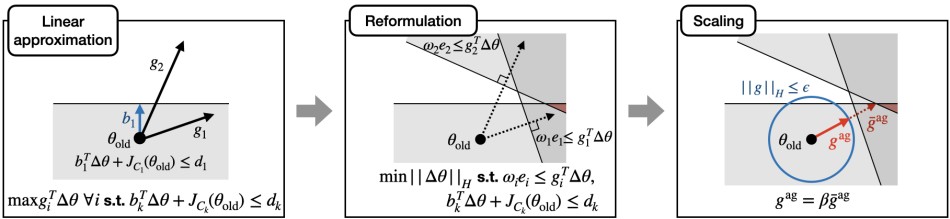

Figure 3: **Process of CoMOGA**. We visualize the process in the parameter space, and the gray areas represent constraints. **(Linear approximation)** CoMOGA linearly approximates the original CMORL problem in (1). The gradients of the objective and constraint functions are visualized as black and blue arrows, respectively. **(Reformulation)** The objectives are converted to constraints as described in (5). The intersection of all constraints is shown as the red area. **(Scaling)** The solution of the transformed problem, $\bar{g}^{\text{ag}}$, is scaled to ensure that the updated policy remains within the local region.

considering only the safety constraints when the current policy violates them, which is motivated by safe RL algorithms (Kim et al., 2023; Xu et al., 2021; Zhang et al., 2022). We will show that this step helps the policy converge to global optima in Section 4.4. The recovery step involves solving the following problem, which solely consists of the safety constraints:

$$\bar{g}_\omega^{\text{ag}} := \operatorname{argmin}_{\Delta\theta} \Delta\theta^T H \Delta\theta \quad \text{s.t. } b_k^T \Delta\theta + J_{C_k}(\theta_{\text{old}}) \le d_k \ \forall k. \tag{6}$$

As a result, the process for obtaining the policy gradient is depicted in Figure 3 and can be formalized as follows, which is named *constrained multi-objective gradient aggregator (CoMOGA)*:

$$\bar{g}_\omega^{\text{ag}} = \begin{cases} \operatorname{argmin}_{\Delta\theta} \Delta\theta^T H \Delta\theta \ \text{s.t. } \omega_i e_i \le g_i^T \Delta\theta, \ b_k^T \Delta\theta + J_{C_k}(\theta_{\text{old}}) \le d_k & \text{if } J_{C_k}(\theta_{\text{old}}) \le d_k \ \forall k, \\ \operatorname{argmin}_{\Delta\theta} \Delta\theta^T H \Delta\theta \ \text{s.t. } b_k^T \Delta\theta + J_{C_k}(\theta_{\text{old}}) \le d_k & \text{otherwise,} \end{cases}$$

$$g_\omega^{\text{ag}} = \min(1, \epsilon/||\bar{g}_\omega^{\text{ag}}||_H)\bar{g}_\omega^{\text{ag}}.$$

We also provide a detailed explanation of the differences between CoMOGA and existing conflict-averse MTL algorithms in Appendix E.

## 4.3 UPDATE RULE FOR UNIVERSAL POLICY

We have introduced the gradient aggregation method named *CoMOGA* for updating a policy for a given single preference. This section presents a method for updating a universal policy that can cover the entire preference space. If a preference $^1\omega$ is given, the policy should be updated to $\theta + g_{1_\omega}^{\text{ag}}$. Similarly, if the preference is $^2\omega$, the policy should be updated to $\theta + g_{2_\omega}^{\text{ag}}$. Consequently, the ideal universal policy is required to satisfy the following conditions:

$$\pi_{\theta_{\text{new}}}(a|s,\omega) = \pi_{\theta_{\text{old}}+g_\omega^{\text{ag}}}(a|s,\omega) \quad \forall a \in A, \ \forall s \in S, \text{ and } \forall \omega \in \Omega,$$

where we denote $\pi_{\theta_{\text{old}}+g_\omega^{\text{ag}}}$ as an *intermediate policy* for $\omega$. This can be interpreted as the KL divergence between the universal policy and the intermediate policies being zero. Based on this property, we introduce a practical method to achieve the universal policy by minimizing the following loss:

$$\min_\theta \mathbb{E}_{(\tau,\omega)\sim\mathcal{D}} \left[ \mathbb{E}_{s\sim\tau} \left[ D_{\text{KL}}(\pi_{\theta_{\text{old}}+g_\omega^{\text{ag}}}(\cdot|s,\omega)||\pi_\theta(\cdot|s,\omega)) \right] \right], \tag{7}$$

where $\mathcal{D}$ is a replay buffer, and $\tau$ is a trajectory. By minimizing the above loss, we can combine policies for each preference into a single universal policy. After training is completed, users can utilize diverse policies by providing personalized preferences to the trained universal policy, eliminating the need for additional training.

## 4.4 CONVERGENCE ANALYSIS

In this section, we analyze the convergence properties of the proposed method. Specifically, it can be shown that CoMOGA achieves CP optimal convergence with minor modifications if $H$ is the Fisher information matrix. To this end, we first introduce the necessary conditions for the policy gradient method to ensure CP optimal convergence. We then show that the modified CoMOGA satisfies these necessary conditions. Before that, we first introduce the following assumption:

**Assumption 4.1** (Slater's condition). *There exists a feasible policy $\pi_f$ such that $J_{C_k}(\pi_f) \le d_k - \eta$ $\forall k$, where $\eta > 0$ is a positive number.*

The Slater's condition is a common assumption in safe RL to ensure optimal convergence (Kim et al., 2023; Bai et al., 2022; Ding et al., 2022). We also assume that the state and action spaces are finite. Now, we introduce a generalized version of the policy gradient method that guarantees optimal convergence.

**Theorem 4.2.** *Assume that sequences* $\{\nu_{t,i}^a, \nu_{t,i}^b, \lambda_{t,k}^a, \lambda_{t,k}^b\}_{t=1}^{\infty} \subseteq [0, \lambda_{\max}]$ *are given for all* $i$ *and* $k$, *where* $\sum_k \lambda_{t,k}^b = 1$, $\lambda_{t,k}^b(J_{C_k}(\theta_t) - d_k) \geq 0$, $\sum_i \nu_{t,i}^a = 1$, *and* $\nu_{t,i}^a$ *converges to a specific point* $\bar{\nu}_i^a$. *Then, if a policy is updated by the following rule, it converges to a CP optimal policy of (1):*

$$\theta_{t+1} = \theta_t + \begin{cases} \alpha_t F^{\dagger}(\theta_t) \left( \sum_{i=1}^N \nu_{t,i}^a g_i - \alpha_t \sum_{k=1}^M \lambda_{t,k}^a b_k \right) & \text{if } J_{C_k}(\theta_t) \leq d_k \ \forall k, \\ \alpha_t F^{\dagger}(\theta_t) \left( \alpha_t \sum_{i=1}^N \nu_{t,i}^b g_i - \sum_{k=1}^M \lambda_{t,k}^b b_k \right) & \text{otherwise,} \end{cases} \quad (8)$$

*where* $g_i = \nabla J_{R_i}(\theta_t)$, $b_k = \nabla J_{C_k}(\theta_t)$, $\alpha_t$ *is a step size satisfying Robbins-Monro condition (Robbins & Monro, 1951), and* $F^{\dagger}$ *represents the pseudo-inverse of the Fisher information matrix.*

The proof is provided in Appendix B.2. The sequences $\nu_{t,i}^a$, $\nu_{t,i}^b$, $\lambda_{t,k}^a$, and $\lambda_{t,k}^b$ in Theorem 4.2 represent the weights assigned to the gradients of the objective and constraint functions at each update step, and a new CMORL algorithm can be developed based on how the sequences are set. Similarly, we will show that CoMOGA can be formulated as (8) by identifying these sequences with minor modifications. Since the quadratic programming holds strong duality, we can express the solution of (5) as $\bar{g}_{\omega}^{\text{ag}} = H^{-1}(\sum_{i=1}^N \nu_i^* g_i - \sum_{k=1}^M \lambda_k^* b_k)$, where $\nu_i^*$ and $\lambda_k^*$ are the optimal Lagrange multipliers of the following dual problem:

$$\nu^*, \lambda^* = \underset{\nu \geq 0, \lambda \geq 0}{\operatorname{argmax}} -\frac{1}{2} \left( \sum_i \nu_i g_i - \sum_k \lambda_k b_k \right)^T H^{-1} \left( \sum_i \nu_i g_i - \sum_k \lambda_k b_k \right) + \sum_i \nu_i \omega_i e_i$$
$$+ \sum_k \lambda_k (J_{C_k}(\theta_{\text{old}}) - d_k).$$

In case of the dual problem has no solution, we set $\nu_i^*$ and $\lambda_k^*$ to be $\bar{g}_{\omega}^{\text{ag}} = 0$. For the solution of (6), the same process is applied to achieve $\lambda_k^*$. Then, the CoMOGA is modified as follows:

$$\bar{g}_{\omega}^{\text{ag}} = H^{-1} \cdot \begin{cases} \sum_{i=1}^N \frac{\nu_i^*}{\sum_j \nu_j^*} g_i - \epsilon_t \sum_{k=1}^M \min(\frac{\lambda_k^*}{\epsilon_t \sum_j \nu_j^*}, \lambda_{\max}) b_k & \text{if } J_{C_k}(\theta_t) \leq d_k \ \forall k, \\ -\sum_{k=1}^M (\lambda_k^*/(\sum_j \lambda_j^*)) b_k & \text{otherwise,} \end{cases} \quad (9)$$

$$g_{\omega}^{\text{ag}} = \epsilon_t \bar{g}_{\omega}^{\text{ag}} / \min(\max(\|\bar{g}_{\omega}^{\text{ag}}\|_H, g_{\min}), g_{\max}),$$

where $g_{\min}$, $g_{\max}$, $\lambda_{\max} \in \mathbb{R}_{\geq 0}$ are hyperparameters, and $\epsilon_t$ is the local region size satisfying Robbins-Monro condition. Finally, we can show optimal convergence of the modified CoMOGA.

**Theorem 4.3.** *If* $H$ *is the Fisher information matrix and a policy* $\pi_{\theta_t}$ *is updated as* $\theta_{t+1} = \theta_t + g_{\omega}^{\text{ag}}$, *where* $g_{\omega}^{\text{ag}}$ *is defined in (9), it converges to a CP optimal policy of (1).*

Theorem 4.3 can be proved by identifying sequences $\nu_{t,i}^a$, $\nu_{t,k}^b$, $\lambda_{t,i}^a$, $\lambda_{t,k}^b$ that satisfy the conditions mentioned in Theorem 4.2, and details are provided in Appendix B.3.

## 5 PRACTICAL IMPLEMENTATION

We need to compute the gradient of the objective and constraint functions to perform CoMOGA and the universal policy update. To do that, we use reward and cost critics $V_{R,\psi_R}^{\pi}$, $V_{C,\psi_C}^{\pi}$, where $\psi_R$ and $\psi_C$ are the critic network parameters, to estimate the objective and constraint functions. The critics are trained by minimizing the following loss functions:

$$\mathcal{L}(\psi_R) := \underset{(s,a,s',\omega) \sim \mathcal{D}}{\mathbb{E}} \left[ \sum_{i=1}^N (Y_{R_i} - V_{R_i,\psi_R}^{\pi}(s,a,\omega))^2 \right], \quad (10)$$

where the target $Y_{R_i}$ can be calculated either as $R_i(s,a,s') + \gamma V_{R_i,\psi_R}^{\pi}(s',a',\omega)$ with $a' \sim \pi(\cdot|s',\omega)$ or using Retrace($\lambda$) (Munos et al., 2016), and the same process is applied to the cost critics. Given a preference $\omega$, the objective and constraint functions can be estimated as follows:

$$J_{R_i}(\theta) \approx \underset{\substack{s \sim \mathcal{D}, \\ a \sim \pi_{\theta}(\cdot|s,\omega)}}{\mathbb{E}} \left[ V_{R_i,\psi_R}^{\pi}(s,a,\omega) \right], \ J_{C_k}(\theta) \approx \underset{\substack{s \sim \mathcal{D}, \\ a \sim \pi_{\theta}(\cdot|s,\omega)}}{\mathbb{E}} \left[ V_{C_k,\psi_C}^{\pi}(s,a,\omega) \right]. \quad (11)$$

Then, we can compute the policy gradients using the critic networks through the reparameterization trick, as done in the SAC paper (Haarnoja et al., 2018). For the details of solving COP, please refer to Appendix C.3.1. Finally, the proposed method is summarized in Algorithm 1.

---

**Algorithm 1** Policy Update Using CoMOGA

---

**Input:** Policy parameter $\theta$, reward critic parameter $\psi_R$, cost critic parameter $\psi_C$, replay buffer $\mathcal{D}$.
**for** epochs $= 1$ **to** $E$ **do**
    **for** rollouts $= 1$ **to** $L$ **do**
        Sample a preference $\omega \sim \Omega$.
        Collect a trajectory $\tau = \{(s_t, a_t, r_t, c_t, s_{t+1})\}_{t=1}^{T}$ by using $\pi_\theta(\cdot|\cdot, \omega)$ and store $(\tau, \omega)$ in $\mathcal{D}$.
    **end for**
    Update reward and cost critic networks using Equation (10).
    **for** $p = 1$ **to** $P$ **do**
        Sample $(\tau, \omega) \sim D$, and estimate the objectives and constraints using Equation (11).
        Calculate the aggregated gradient $g_\omega^{\mathrm{ag}}$ using Equation (9), and set $\theta_\omega = \theta_{\mathrm{old}} + g_\omega^{\mathrm{ag}}$.
        Store an intermediate policy $\pi_{\theta_\omega}$.
    **end for**
    Update the universal policy using Equation (7).
**end for**

---

## 6 EXPERIMENTS

This section evaluates the proposed method and baselines across various tasks with and without constraints. First, we explain how methods are evaluated on the tasks and then present the CMORL baselines. Subsequently, each task is described, and the results are analyzed. Finally, we conclude this section with ablation studies of the proposed method.

### 6.1 EVALUATION METRICS AND BASELINES

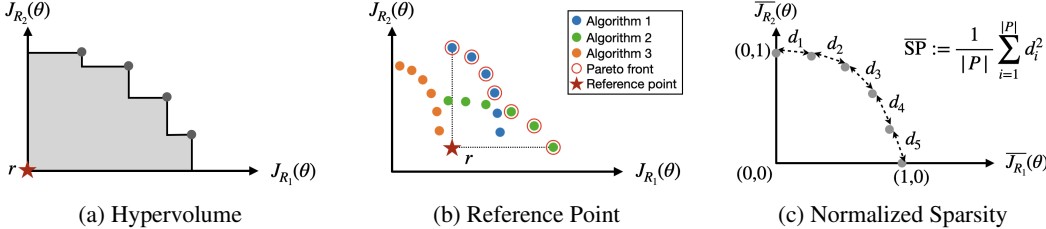

(a) Hypervolume        (b) Reference Point        (c) Normalized Sparsity

Figure 4: **Visualization of metrics.** In (a), the hypervolume is represented by the gray area. In (b), the red star indicates the reference point, determined from the entire Pareto front, whose elements are circled in red. In (c), the normalized sparsity is calculated by averaging the squared distances between elements in the normalized space.

**Metrics.** Once the universal policy is trained, an estimated CP front, denoted as $P \subset \mathbb{R}^N$, can be obtained by calculating the objective values $(J_{R_1}(\pi), ..., J_{R_N}(\pi))$ for a range of preferences. Given that CMORL aims to approximate the ground truth CP front, it is essential to assess how close the estimated CP front is to the ground truth, and it can be realized by measuring the coverage and density of the estimated front. To this end, we use the *hypervolume* (HV) for coverage measurement and *sparsity* (SP) for density estimation, which are commonly used in many MORL algorithms (Xu et al., 2020; Basaklar et al., 2023). The visualization of metrics is shown in Figure 4, and detailed descriptions are provided in Appendix C.2. Given an estimated CP front $P \subset \mathbb{R}^N$ and a reference point $r \in \mathbb{R}^N$, the hypervolume (HV) is defined as follows (Zitzler & Thiele, 1999):

$$\mathrm{HV}(P, r) := \int_{\mathbb{R}^N} \mathbf{1}_{Q(P,r)}(z)dz, \tag{12}$$

where $Q(P, r) := \{z | \exists p \in P \text{ s.t. } r \preceq z \preceq p\}$, and the metric represents the volume of the area surrounding the reference point and the estimated CP front. Sparsity (SP), on the other hand, measures how uniformly the estimated CP fronts are distributed and is defined as the average of the squared distances between elements of the CP fronts (Xu et al., 2020). However, the original definition exhibits a correlation with HV, where a larger HV corresponds to a larger SP. To eliminate this inherent correlation, we propose to use the following normalized version of SP:

$$\overline{\mathrm{SP}}(P) := \frac{1}{|P|-1} \sum_{j=1}^{N} \sum_{i=1}^{|P|-1} \left( \frac{\tilde{P}_j[i] - \tilde{P}_j[i+1]}{\max_k \tilde{P}_j[k] - \min_k \tilde{P}_j[k]} \right)^2, \tag{13}$$

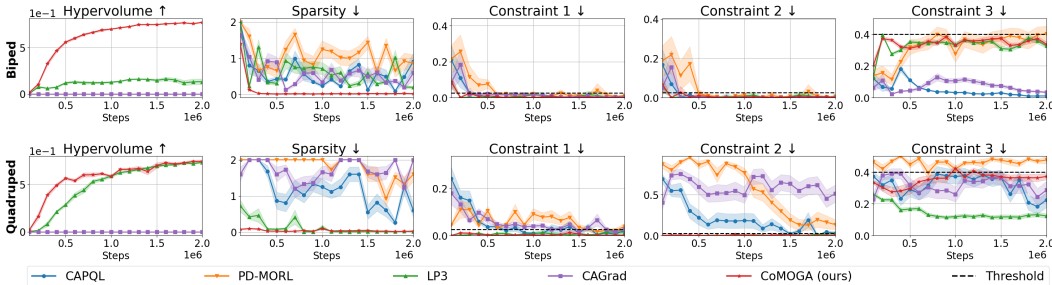

Figure 5: Evaluation results of the legged robot locomotion tasks. The upper row shows results for the bipedal robot, while the lower row is for the quadrupedal robot. All algorithms are evaluated at every $10^5$ steps. The bold lines and shaded areas represent the mean and quarter-scaled standard deviation of results from five random seeds, respectively. The black dotted lines in constraint graphs indicate the thresholds.

where $\tilde{P}_j := \mathrm{Sort}(\{p[j]|\forall p \in P\})$, and $\tilde{P}_j[i]$ is the $i$th element in $\tilde{P}_j$. By normalizing the distance using the minimum and maximum values of the CP fronts, we can remove the correlation with HV while preserving the ability to measure sparsity. Additionally, to reduce the influence of the scale of each objective, we normalize each objective within the CP front using the minimum and maximum values from the entire dataset.

**Baselines.** For comparison with the proposed method, we first employ *LP3* (Huang et al., 2022), an existing CMORL algorithm, as a baseline. This method extends the scale-invariant MORL algorithm, *MOMPO* (Abdolmaleki et al., 2020), to CMORL by handling constraints in a similar way to the Lagrangian method. To provide more comprehensive baselines, we include MORL algorithms that are based on policy gradient approaches, which can be adapted to handle constraints using the Lagrangian method. There are two state-of-the-art policy gradient-based MORL algorithms: *PD-MORL* (Basaklar et al., 2023), which is based on TD3 (Fujimoto et al., 2018), and *CAPQL* (Lu et al., 2023), which is based on SAC (Haarnoja et al., 2018). Also, we include conflict-averse MTL algorithms, *CAGrad* (Liu et al., 2021). We have extended them to CMORL by handling constraints using the Lagrangian method. For implementation details, please refer to Appendix C.3.2.

## 6.2 LEGGED ROBOT LOCOMOTION

The legged robot locomotion tasks (Kim et al., 2023) are to control a quadrupedal or bipedal robot to follow randomly given commands while satisfying three constraints: keeping 1) body balance, 2) CoM height, and 3) pre-defined foot contact timing. Since the original tasks were designed to have a single objective, we have modified the tasks to have two objectives: 1) matching the current velocity with the command and 2) minimizing energy consumption. Please see Appendix C for more details.

The evaluation results are presented in Figure 5, and the estimated CP fronts are visualized in Figure 7. Across all tasks, CoMOGA achieves the highest HV and the lowest SP while satisfying all constraints. Notably, CoMOGA surpasses the baselines in the bipedal task. This suggests that preventing gradient conflicts is particularly effective for dynamically unstable robots, such as bipedal robots, because these robots require a careful balance of multiple objectives to attain desired behaviors. LP3 shows performance similar to the proposed method in the quadrupedal task but underperforms in the bipedal task.

## 6.3 SAFETY GYMNASIUM

We utilize single-agent and multi-agent goal tasks in the Safety Gymnasium (Ji et al., 2023). These tasks require navigating robots to designated goals while avoiding obstacles. The single-agent goal tasks have two objectives: 1) reaching goals as many times as possible within a time limit and 2) maximizing energy efficiency, along with a single constraint to avoid collisions with obstacles. The multi-agent goal tasks have two objectives and two constraints, which are to maximize goal achievement and avoid collisions for two robots, respectively. In both tasks, point and car robots are used. For details on the tasks and hyperparameter settings, please refer to Appendix C.

The training curves are presented in Figure 6, and the estimated CP fronts are shown in Figure 7. CoMOGA shows the highest HV in all tasks and the lowest SP in two out of four tasks, while

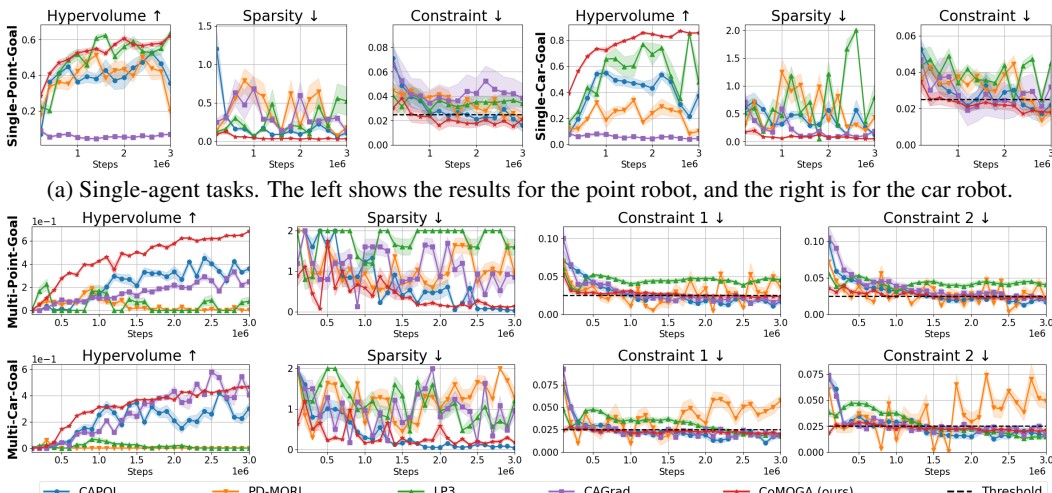

(a) Single-agent tasks. The left shows the results for the point robot, and the right is for the car robot.

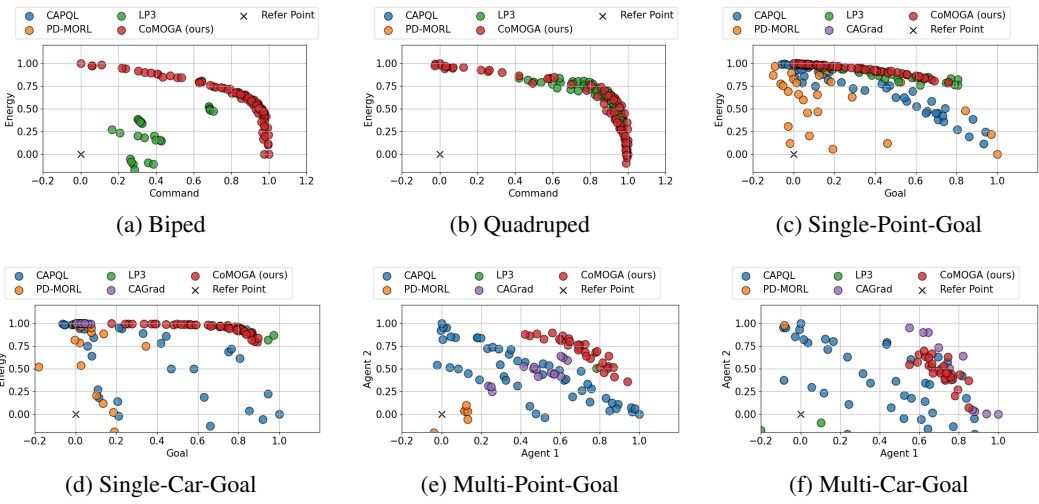

(b) Multi-agent tasks. The upper row shows the results for the point robot, and the lower is for the car robot.

Figure 6: Evaluation results of the Safety Gymnasium tasks.

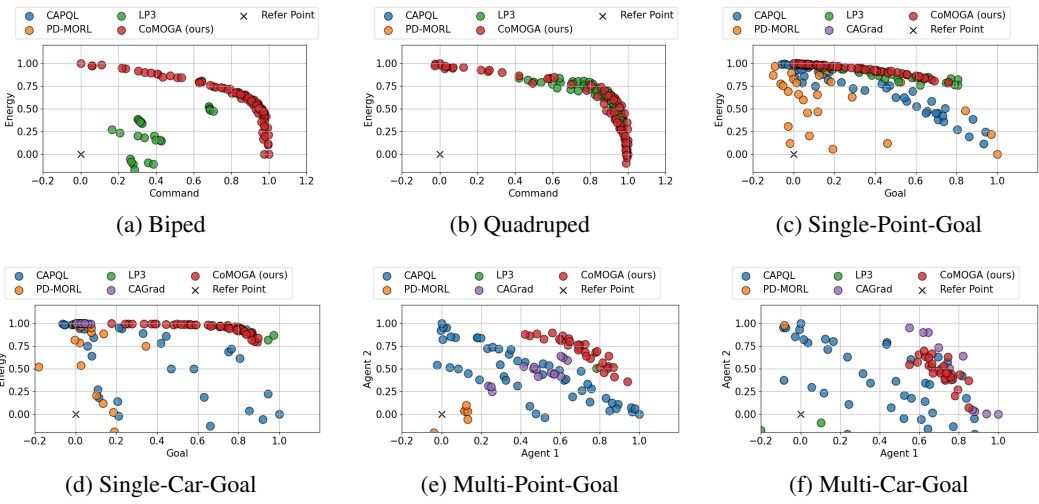

(a) Biped      (b) Quadruped      (c) Single-Point-Goal

(d) Single-Car-Goal      (e) Multi-Point-Goal      (f) Multi-Car-Goal

Figure 7: Visualization of estimated CP fronts for CMORL tasks.

satisfying all constraints. Additionally, the low volatility in the evaluation graph indicates that the proposed method can stably handle both constraints and objectives. LP3 and PD-MORL violate constraints in at least one task, and CAPQL satisfies all constraints but shows weak performance in the single agent tasks, implying the difficulty of handling constraints and objectives simultaneously. CAGrad shows a similar HV in the Multi-Car-Goal task, but it has a large sparsity.

## 6.4 MULTI-OBJECTIVE GYMNASIUM

We conduct experiments in the Multi-Objective (MO) Gymnasium (Alegre et al., 2022), which is a well-known MORL environment, to examine whether the proposed method also performs well on unconstrained MORL tasks. We use the MuJoCo tasks with continuous action spaces in the MO Gymnasium, and there are six tasks available: Hopper, Humanoid, Half-Cheetah, Walker2d, Ant, and Swimmer. Each task has three, two, two, two, three and two objectives, respectively. Details are provided in Appendix C.

The training curves are presented in Figure 8, the estimated Pareto fronts are detailed in Appendix D. The proposed method achieves the highest HV in three tasks and the lowest SP in five out of six tasks. Especially, CoMOGA exhibits outstanding performance in the hopper and humanoid tasks, which suggests that the conflict-averse strategy is effective for dynamically unstable robots even in

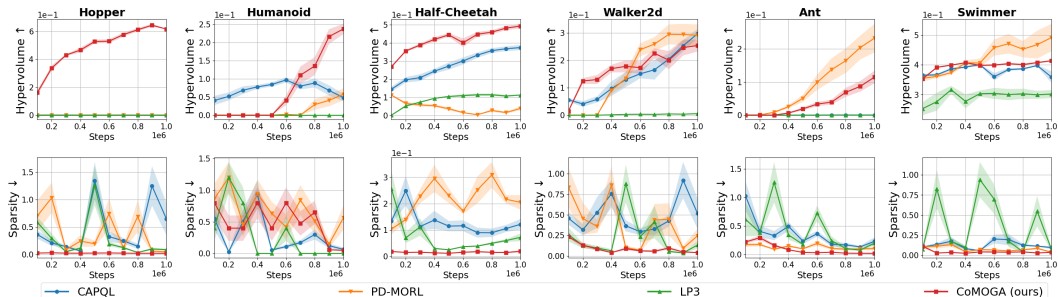

Figure 8: Evaluation results of the MO Gymnasium tasks. Each column shows the results of the task corresponding to its title.

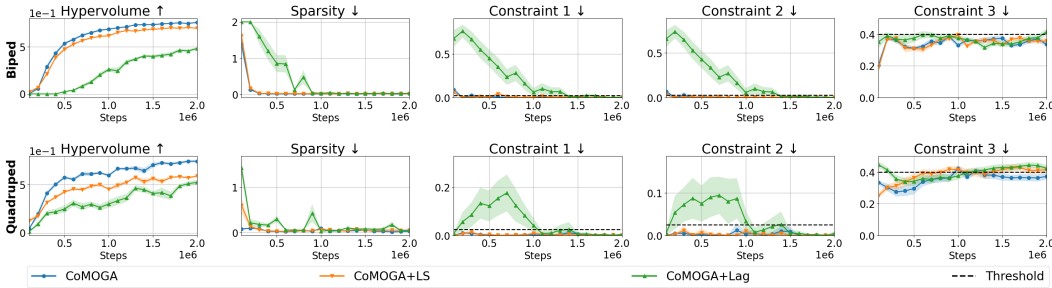

Figure 9: Results of the ablation study in the legged robot locomotion tasks.

unconstrained MORL settings. However, all algorithms except LP3 show high HV results in the walker2d task, which is also dynamically unstable. It can be considered that they have reached the global optimum due to the well-designed reward functions.

### 6.5 ABLATION STUDY

This section evaluate the effectiveness of the core components of the proposed method. The core components include **(1)** transforming objectives into constraints to avoid gradient conflicts and **(2)** handling safety constraints by solving a QP problem, where the constraints are linearly approximated within the local region. To evaluate the efficiency of these components, we designed two ablation methods: 1) *CoMOGA+LS*, which excludes (1) and uses linear scalarization to handle multiple objectives, and 2) *CoMOGA+Lag*, which excludes (2) and handles constraints using the Lagrangian method. The training results on the legged locomotion tasks are presented in Figure 9. The results indicate that CoMOGA+Lag suffers from instability in handling safety constraints due to the use of Lagrange multipliers. Meanwhile, CoMOGA+LS ensures stable satisfaction of safety constraints but achieves lower HVs compared to CoMOGA, confirming the effectiveness of CoMOGA in avoiding gradient conflicts. Additionally, we present the training results for the quadruped task across various hyperparameter values in Appendix D to evaluate sensitivity.

### 7 CONCLUSIONS AND LIMITATIONS

We have introduced a CMORL algorithm called CoMOGA to maximize multiple objectives while satisfying safety constraints. The proposed method is based on a novel reformulation process that converts objectives into constraints, which enables the avoidance of gradient conflicts among multiple objectives and the handling of constraints without additional optimization variables. We show that the proposed method converges to a CP optimal policy in tabular settings and demonstrate that it achieves outstanding performance in HV and SP metrics with constraint satisfaction through various experiments. Since the proposed method updates the policy in a manner that prevents any objective functions from decreasing, the set of trained policies may cover a narrow portion of the CP front. Future research can address this limitation by mitigating the condition of gradient conflicts to better balance multiple objectives and expand coverage. Specifically, by satisfying the conditions of the proposed generalized policy update rule in Theorem 4.2, various CMORL algorithms can be developed with convergence guarantees, which can accelerate future work.

ACKNOWLEDGMENTS

This work was supported by Institute of Information & Communications Technology Planning & Evaluation (IITP) grant funded by the Korea government (MSIT) (No. 2019-0-01190, [SW Star Lab] Robot Learning: Efficient, Safe, and Socially-Acceptable Machine Learning).

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

# A  TOY EXAMPLE

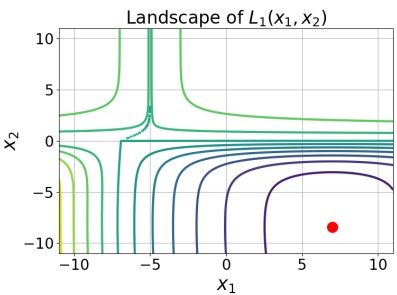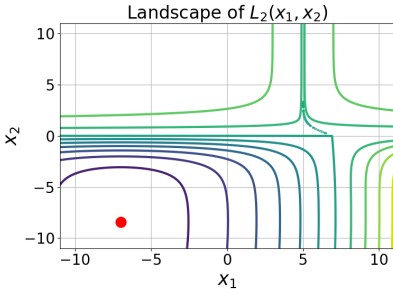

Figure 10: Landscape of the objective functions of the toy example.

In this section, we describe the details of the toy example, which is initially defined in (Liu et al., 2021). Objective functions of the toy example are formulated as follows:

$$
\begin{aligned}
L_1(x_1, x_2) &:= c_1(x_2)f_1(x_1, x_2) + c_2(x_2)g_1(x_1, x_2), \\
L_2(x_1, x_2) &:= c_1(x_2)f_2(x_1, x_2) + c_2(x_2)g_2(x_1, x_2), \text{ where} \\
c_1(x) &:= \max(\tanh(0.5x), 0), \\
c_2(x) &:= \max(\tanh(-0.5x), 0), \\
f_1(x_1, x_2) &:= \log(\max(|0.5(-x_1 - 7) - \tanh(-x_2)|, 0.000005)) + 6, \\
f_2(x_1, x_2) &:= \log(\max(|0.5(-x_1 + 3) + \tanh(-x_2 + 2)|, 0.000005)) + 6, \\
g_1(x_1, x_2) &:= ((-x_1 + 7)^2 + 0.1(-x_2 - 8)^2)/10 - 20, \\
g_2(x_1, x_2) &:= ((-x_1 - 7)^2 + 0.1(-x_2 - 8)^2)/10 - 20,
\end{aligned}
$$

where $L_1$ and $L_2$ are the objective functions. The landscape of the objective functions are presented in Figure 10. In addition, we define a constraint function, which is defined as follows:

$$
C(x_1, x_2) := x_1^2 + 0.3(x_2 - 10)^2 - 10.5^2.
$$

Finally, the toy example is formulated as follows:

$$
\min_{x_1, x_2} L_1(x_1, x_2) \text{ and } L_2(x_1, x_2) \quad \textbf{s.t. } C(x_1, x_2) \leq 0.
$$

We set the preference as $(0.5, 0.5)$ for both algorithms, LS and CoMOGA, and use the following four initial points: $(-10, 0)$, $(-10, 7.5)$, $(0, 7.5)$, and $(10, 10)$. The optimization trajectories for each method are shown in Figure 2. For the initial point $(-10, 7.5)$, the gradient of $L_1$ is much larger than $L_2$. Thus, addressing the toy example with LS can be significantly influenced by $L_1$ and eventually converges to a local optimal point of $L_1$. To resolve this issue, avoiding gradient conflicts can prevent policy gradients from being dominated by a specific objective, which helps them converge to an optimal point.

# B PROOFS

## B.1 PROOF OF REFORMULATION PROCESS

In this section, we prove that the solutions of (2) and (3) are equivalent. Under the linear assumption, we can express the $i$th objective function as $g_i^T \Delta\theta + J_{R_i}(\theta_{\text{old}})$. Then, (2) can be rewritten as follows:

$$\max_{\Delta\theta} \ g_i^T \Delta\theta + J_{R_i}(\theta_{\text{old}}) \ \text{ s.t. } \Delta\theta^T H \Delta\theta \le \epsilon^2.$$

As the strong duality holds, we will find the solution by solving the following Lagrange dual problem:

$$\max_{\lambda \ge 0} \min_{\Delta\theta} \ -g_i^T \Delta\theta + \lambda(\Delta\theta^T H \Delta\theta - \epsilon^2) =: L(\Delta\theta, \lambda).$$

Then, $\Delta\theta^*(\lambda) = \text{argmin}_{\Delta\theta} L(\Delta\theta, \lambda) = H^{-1} g_i/(2\lambda)$. By replacing $\Delta\theta$ with $\Delta\theta^*(\lambda)$ in the dual problem, we get $\lambda^* = \text{argmax}_\lambda L(\Delta\theta^*(\lambda), \lambda) = \sqrt{g_i^T H^{-1} g_i}/(2\epsilon)$. Finally, we obtain the solution, $\Delta\theta^* = \Delta\theta^*(\lambda^*) = \epsilon H^{-1} g_i/\sqrt{g_i^T H^{-1} g_i}$.

Now let us find the solution of (3). Under the linear assumption, we can rewrite (3) as follows:

$$\min_{\Delta\theta} \ \Delta\theta^T H \Delta\theta \ \text{ s.t. } e_i \le g_i^T \Delta\theta. \tag{14}$$

Similar to the above process, the Lagrange dual problem is derived as follows:

$$\max_{\lambda \ge 0} \min_{\Delta\theta} \ \Delta\theta^T H \Delta\theta + \lambda(e_i - g_i^T \Delta\theta) =: L(\Delta\theta, \lambda). \tag{15}$$

Then, the solution of the dual problem is obtained as $\Delta\theta^*(\lambda) = \text{argmin}_{\Delta\theta} L(\Delta\theta, \lambda) = \lambda H^{-1} g_i/2$. Using $\Delta\theta^*(\lambda)$, we can get the optimal Lagrange multiplier as $\lambda^* = \text{argmax}_\lambda L(\Delta\theta^*(\lambda), \lambda) = 2e_i/(g_i H^{-1} g_i) = 2\epsilon/\sqrt{g_i H^{-1} g_i}$. Then, the solution is $\Delta\theta^* = \Delta\theta^*(\lambda^*) = \epsilon H^{-1} g_i/\sqrt{g_i^T H^{-1} g_i}$. As a result, the solutions of (2) and (3) are the same as $\epsilon H^{-1} g_i/\sqrt{g_i^T H^{-1} g_i}$.

## B.2 PROOF OF THEOREM 4.2

We first define the following notions for a policy $\pi$:

$$d_\rho^\pi(s) := (1-\gamma) \sum_{t=0}^\infty \gamma^t \text{Pr}(s_t = s) \Big| s_0 \sim \rho, a_t \sim \pi(\cdot|s_t), s_{t+1} \sim P(\cdot|s_t, a_t) \ \forall t,$$

$$V^\pi(s) := \mathbb{E}\left[\sum_{t=0}^\infty \gamma^t R(s_t, a_t, s_{t+1}) \Big| s_0 = s, a_t \sim \pi(\cdot|s_t), s_{t+1} \sim P(\cdot|s_t, a_t) \ \forall t\right],$$

$$Q^\pi(s,a) := \mathbb{E}\left[\sum_{t=0}^\infty \gamma^t R(s_t, a_t, s_{t+1}) \Big| s_0 = s, a_0 = a, s_{t+1} \sim P(\cdot|s_t, a_t), a_{t+1} \sim \pi(\cdot|s_{t+1}) \ \forall t\right],$$

$$A^\pi(s,a) := Q^\pi(s,a) - V^\pi(s).$$

We also simplify several notations for brevity as follows:

$$\pi_{\theta_t} \to \pi_t, \ d_\rho^{\pi_{\theta_t}} \to d_\rho^t, \ A_{R_i}^{\pi_{\theta_t}}(s,a) \to A_{R_i}^t(s,a), \ A_{C_k}^{\pi_{\theta_t}}(s,a) \to A_{C_k}^t(s,a).$$

Since it is assumed that the state and action spaces are finite, a policy can be parameterized using a softmax parameterization as follows (Agarwal et al., 2021):

$$\pi_\theta(a|s) := \exp(\theta(s,a)) \Big/ \sum_{a' \in A} \exp(\theta(s,a')),$$

where $\theta \in \mathbb{R}^{S \times A}$ is a trainable parameter, and $\theta(s,a)$ denotes the value corresponding to state $s$ and action $a$. Through Lemma 4 in (Xu et al., 2021), the following equations are satisfied when a policy is updated according to (8):

$$\theta_{t+1} = \theta_t(s,a) + \frac{\alpha_t}{1-\gamma} \cdot \begin{cases} \left(\sum_{i=1}^N \nu_{t,i}^a A_{R_i}^t - \alpha_t \sum_{k=1}^M \lambda_{t,k}^a A_{C_k}^t\right) & \text{if } J_{C_k}(\theta_t) \le d_k \ \forall k, \\ \left(\alpha_t \sum_{i=1}^N \nu_{t,i}^b A_{R_i}^t - \sum_{k=1}^M \lambda_{t,k}^b A_{C_k}^t\right) & \text{otherwise}, \end{cases}$$

$$\pi_{t+1}(a|s) = \pi_t(a|s) \frac{\exp(\theta_{t+1}(s,a) - \theta_t(s,a))}{Z_t(s)},$$

where $Z_t(s) := \sum_{a \in A} \pi_t(a|s) \exp(\theta_{t+1}(s,a) - \theta_t(s,a)).$

Now, we introduce a lemma showing that the LS approach guarantees to converge a CP optimal policy, followed by the proof of Theorem 4.3.

**Lemma B.1.** *Given a preference $\omega$, let us define an LS optimal policy as follows:*

$$\pi_\omega^* := \arg\max_\pi \sum_{i=1}^N \omega_i J_{R_i}(\pi) \quad \text{s.t. } J_{C_k}(\pi) \le d_k \; \forall k.$$

*Then, the LS optimal policy is also a CP optimal policy.*

*Proof.* Let us assume that there exists a policy $\mu$ that dominates the LS optimal policy $\pi_\omega^*$. By definition of the constrained dominance, the following is satisfied:

$$J_{R_i}(\mu) > J_{R_i}(\pi_\omega^*) \; \forall i \text{ and } J_{C_k}(\mu) \le d_k \; \forall k. \Rightarrow \sum_{i=1}^N \omega_i(J_{R_i}(\mu) - J_{R_i}(\pi_\omega^*)) > 0.$$

However, this equation contradicts the fact that $\pi_\omega^*$ is an LS optimal policy. Consequently, $\pi_\omega^*$ is not dominated by any others, it is also a CP optimal policy. $\qquad\square$

**Theorem 4.2.** *Assume that sequences $\{\nu_{t,i}^a, \nu_{t,i}^b, \lambda_{t,k}^a, \lambda_{t,k}^b\}_{t=1}^\infty \subseteq [0, \lambda_{\max}]$ are given for all $i$ and $k$, where $\sum_k \lambda_{t,k}^b = 1$, $\lambda_{t,k}^b(J_{C_k}(\theta_t) - d_k) \ge 0$, $\sum_i \nu_{t,i}^a = 1$, and $\nu_{t,i}^a$ converges to a specific point $\bar\nu_i^a$. Then, if a policy is updated by the following rule, it converges to a CP optimal policy of (1):*

$$\theta_{t+1} = \theta_t + \begin{cases} \alpha_t F^\dagger(\theta_t)\left(\sum_{i=1}^N \nu_{t,i}^a g_i - \alpha_t \sum_{k=1}^M \lambda_{t,k}^a b_k\right) & \text{if } J_{C_k}(\theta_t) \le d_k \; \forall k, \\ \alpha_t F^\dagger(\theta_t)\left(\alpha_t \sum_{i=1}^N \nu_{t,i}^b g_i - \sum_{k=1}^M \lambda_{t,k}^b b_k\right) & \text{otherwise,} \end{cases} \tag{8}$$

*where $g_i = \nabla J_{R_i}(\theta_t)$, $b_k = \nabla J_{C_k}(\theta_t)$, $\alpha_t$ is a step size satisfying Robbins-Monro condition (Robbins & Monro, 1951), and $F^\dagger$ represents the pseudo-inverse of the Fisher information matrix.*

*Proof.* We first consider the case $J_{C_k}(\theta_t) \le d_k \; \forall k$, in which the policy is updated as follows:

$$\theta_{t+1} = \theta_t + \frac{\alpha_t}{1-\gamma}\left(\sum_{i=1}^N \nu_{t,i}^a A_{R_i}^t - \alpha_t \sum_{k=1}^M \lambda_{t,k}^a A_{C_k}^t\right).$$

Using Lemma 1 in (Schulman et al., 2015), we can derive the following inequality:

$$\begin{aligned} &\sum_i \nu_{t,i}^a(J_{R_i}(\theta_{t+1}) - J_{R_i}(\theta_t)) - \alpha_t \sum_k \lambda_{t,k}^a(J_{C_k}(\theta_{t+1}) - J_{C_k}(\theta_t)) \\ &= \frac{1}{1-\gamma}\mathbb{E}_{s\sim d_\rho^{t+1}}\left[\mathbb{E}_{a\sim\pi_{t+1}(\cdot|s)}\left[\sum_{i=1}^N \nu_{t,i}^a A_{R_i}^t(s,a) - \alpha_t \sum_{k=1}^M \lambda_{t,k}^a A_{C_k}^t(s,a)\right]\right] \\ &= \frac{1}{\alpha_t}\mathbb{E}_{s\sim d_\rho^{t+1}}\left[\mathbb{E}_{a\sim\pi_{t+1}(\cdot|s)}\left[\theta_{t+1}(s,a) - \theta_t(s,a)\right]\right] \\ &= \frac{1}{\alpha_t}\mathbb{E}_{s\sim d_\rho^{t+1}}\left[\mathbb{E}_{a\sim\pi_{t+1}(\cdot|s)}\left[\log\frac{\pi_{t+1}(a|s)Z_t(s)}{\pi_t(a|s)}\right]\right] \\ &= \frac{1}{\alpha_t}\mathbb{E}_{s\sim d_\rho^{t+1}}\left[D_{\text{KL}}(\pi_{t+1}(\cdot|s)||\pi_t(\cdot|s)) + \log Z_t(s)\right] \\ &\ge \frac{1}{\alpha_t}\mathbb{E}_{s\sim d_\rho^{t+1}}\left[\log Z_t(s)\right] \stackrel{\text{(i)}}{\ge} \frac{1-\gamma}{\alpha_t}\mathbb{E}_{s\sim\rho}\left[\log Z_t(s)\right], \end{aligned} \tag{16}$$

where (i) follows from the fact that $d_\rho^\pi = (1-\gamma)\sum_{t=0}^\infty \gamma^t \Pr(s_t = s) \ge (1-\gamma)\rho(s)$. By Lemma 5 in (Xu et al., 2021),

$$|J_{R_i}(\theta_{t+1}) - J_{R_i}(\theta_t)| \le \frac{2R_{\max}}{1-\gamma}||\theta_{t+1} - \theta_t||_2 \; \forall i,$$

$$|J_{C_k}(\theta_{t+1}) - J_{C_k}(\theta_t)| \le \frac{2R_{\max}}{1-\gamma}||\theta_{t+1} - \theta_t||_2 \; \forall k.$$

$$\Rightarrow \left| \sum_i \nu_{t,i}^a (J_{R_i}(\theta_{t+1}) - J_{R_i}(\theta_t)) - \alpha_t \sum_k \lambda_{t,k}^a (J_{C_k}(\theta_{t+1}) - J_{C_k}(\theta_t)) \right|$$

$$\leq \frac{2R_{\max}}{1-\gamma}(N + \alpha_t M)\lambda_{\max}||\theta_{t+1} - \theta_t||_2$$

$$= \frac{2R_{\max}}{1-\gamma}(N + \alpha_t M)\lambda_{\max}\left\| \frac{\alpha_t}{1-\gamma}(\sum_{i=1}^N \nu_{t,i}^a A_{R_i}^t - \alpha_t \sum_{k=1}^M \lambda_{t,k}^a A_{C_k}^t) \right\|_2$$

$$\leq \frac{2\alpha_t R_{\max}^2}{(1-\gamma)^3}(N + \alpha_t M)^2 \lambda_{\max}^2 \sqrt{|S||A|}.$$

Then, (16) can be rewritten as follows:

$$\frac{2\alpha_t R_{\max}^2}{(1-\gamma)^3}(N + \alpha_t M)^2 \lambda_{\max}^2 \sqrt{|S||A|} \geq \frac{1-\gamma}{\alpha_t} \mathbb{E}_{s\sim\rho}\left[\log Z_t(s)\right], \tag{17}$$

which is satisfied for any $\rho$. Finally, we derive the following inequality for a policy $\mu$ as follows:

$$\sum_i \nu_{t,i}^a (J_{R_i}(\mu) - J_{R_i}(\theta_t)) - \alpha_t \sum_k \lambda_{t,k}^a (J_{C_k}(\mu) - J_{C_k}(\theta_t))$$

$$= \frac{1}{1-\gamma}\mathbb{E}_{s\sim d_\rho^\mu}\left[\mathbb{E}_{a\sim\mu(\cdot|s)}\left[\sum_{i=1}^N \nu_{t,i}^a A_{R_i}^t(s,a) - \alpha_t \sum_{k=1}^M \lambda_{t,k}^a A_{C_k}^t(s,a)\right]\right]$$

$$= \frac{1}{\alpha_t}\mathbb{E}_{s\sim d_\rho^\mu}\left[\mathbb{E}_{a\sim\mu(\cdot|s)}\left[\log\frac{\pi_{t+1}(a|s)Z_t(s)}{\pi_t(a|s)}\right]\right] \tag{18}$$

$$= \frac{1}{\alpha_t}\mathbb{E}_{s\sim d_\rho^\mu}\left[D_{\mathrm{KL}}(\mu(\cdot|s)||\pi_t(\cdot|s)) - D_{\mathrm{KL}}(\mu(\cdot|s)||\pi_{t+1}(\cdot|s)) + \log Z_t(s)\right]$$

$$\overset{(i)}{\leq} \frac{1}{\alpha_t}\mathbb{E}_{s\sim d_\rho^\mu}\left[D_{\mathrm{KL}}(\mu(\cdot|s)||\pi_t(\cdot|s)) - D_{\mathrm{KL}}(\mu(\cdot|s)||\pi_{t+1}(\cdot|s))\right]$$

$$+ \frac{2\alpha_t R_{\max}^2}{(1-\gamma)^3}(N + \alpha_t M)^2 \lambda_{\max}^2 \sqrt{|S||A|},$$

where (i) is from (17) by replacing $\rho$ with $d_\rho^\mu$. For brevity, we denote $\mathbb{E}_{s\sim d_\rho^\mu}\left[D_{\mathrm{KL}}(\mu(\cdot|s)||\pi_t(\cdot|s))\right]$ as $D_{\mathrm{KL}}(\mu||\pi_t)$. Now, we consider the second case where the constraints are violated. In this case,

$$\theta_{t+1} = \theta_t + \frac{\alpha_t}{1-\gamma}(\alpha_t \sum_{i=1}^N \nu_{t,i}^b A_{R_i}^t - \sum_{k=1}^M \lambda_{t,k}^b A_{C_k}^t),$$

which is symmetrical to the first case, achieved by replacing $\nu_{t,i}^a \rightarrow \alpha_t \nu_{t,i}^b$ and $\alpha_t \lambda_{t,k}^a \rightarrow \lambda_{t,k}^b$. Using this symmetry property, we can obtain the following inequality from (18):

$$\alpha_t \sum_i \nu_{t,i}^b (J_{R_i}(\mu) - J_{R_i}(\theta_t)) - \sum_k \lambda_{t,k}^b (J_{C_k}(\mu) - J_{C_k}(\theta_t))$$

$$\leq \frac{1}{\alpha_t}(D_{\mathrm{KL}}(\mu||\pi_t) - D_{\mathrm{KL}}(\mu||\pi_{t+1})) + \frac{2\alpha_t R_{\max}^2}{(1-\gamma)^3}(\alpha_t N + M)^2 \lambda_{\max}^2 \sqrt{|S||A|}. \tag{19}$$

Now, we define a set of time steps, $\mathcal{N} := \{t|J_{C_k}(\pi_t) \leq d_k \ \forall k\}$. Then, by summing (18) and (19) over several time steps, the following is satisfied for a policy $\mu$:

$$\sum_{t\in\mathcal{N}}\left(\alpha_t \sum_i \nu_{t,i}^a (J_{R_i}(\mu) - J_{R_i}(\theta_t)) - \alpha_t^2 \sum_k \lambda_{t,k}^a (J_{C_k}(\mu) - J_{C_k}(\theta_t))\right)$$

$$+ \sum_{t\notin\mathcal{N}}\left(\alpha_t^2 \sum_i \nu_{t,i}^b (J_{R_i}(\mu) - J_{R_i}(\theta_t)) - \alpha_t \sum_k \lambda_{t,k}^b (J_{C_k}(\mu) - J_{C_k}(\theta_t))\right) \tag{20}$$

$$\leq D_{\mathrm{KL}}(\mu||\pi_0) + \sum_t \alpha_t^2 \underbrace{\frac{2R_{\max}^2}{(1-\gamma)^3}\max(\alpha_0, 1)^2 (N + M)^2 \lambda_{\max}^2 \sqrt{|S||A|}}_{=:\kappa_1}.$$

If $t \notin \mathcal{N}$, $\lambda_{t,k}^b = 0$ for $J_{C_k}(\theta_t) \leq d_k$, which results in $\sum_k \lambda_{t,k}^b (J_{C_k}(\pi_f) - J_{C_k}(\theta_t)) \leq -\eta$. By using this fact and replacing $\mu$ in (20) with $\pi_f$,

$$\sum_{t \in \mathcal{N}} \left( \alpha_t \sum_i \nu_{t,i}^a (J_{R_i}(\pi_f) - J_{R_i}(\theta_t)) - \alpha_t^2 \sum_k \lambda_{t,k}^a (J_{C_k}(\pi_f) - J_{C_k}(\theta_t)) \right)$$

$$+ \sum_{t \notin \mathcal{N}} \left( \alpha_t^2 \sum_i \nu_{t,i}^b (J_{R_i}(\pi_f) - J_{R_i}(\theta_t)) + \alpha_t \eta \right) \leq D_{\mathrm{KL}}(\pi_f \| \pi_0) + \kappa_1 \sum_t \alpha_t^2.$$

$$\Rightarrow \sum_{t \in \mathcal{N}} \left( \alpha_t \sum_i \nu_{t,i}^a (J_{R_i}(\pi_f) - J_{R_i}(\theta_t)) \right) + \sum_{t \notin \mathcal{N}} \alpha_t \eta$$

$$\leq D_{\mathrm{KL}}(\pi_f \| \pi_0) + \kappa_1 \sum_t \alpha_t^2 + \sum_{t \in \mathcal{N}} \alpha_t^2 \sum_k \lambda_{t,k}^a (J_{C_k}(\pi_f) - J_{C_k}(\theta_t))$$

$$- \sum_{t \notin \mathcal{N}} \alpha_t^2 \sum_i \nu_{t,i}^b (J_{R_i}(\pi_f) - J_{R_i}(\theta_t))$$

$$\leq D_{\mathrm{KL}}(\pi_f \| \pi_0) + \sum_t \alpha_t^2 \underbrace{\left( \kappa_1 + \frac{2R_{\max}\lambda_{\max}}{1 - \gamma} \max(N, M) \right)}_{=:\kappa_2}.$$

Since $\sum_{t \notin \mathcal{N}} \alpha_t \eta = \sum_t \alpha_t \eta - \sum_{t \in \mathcal{N}} \alpha_t \eta$,

$$\sum_{t \in \mathcal{N}} \left( \alpha_t \left( \sum_i \nu_{t,i}^a (J_{R_i}(\pi_f) - J_{R_i}(\theta_t)) - \eta \right) \right) + \sum_t \alpha_t \eta \leq D_{\mathrm{KL}}(\pi_f \| \pi_0) + \sum_t \alpha_t^2 \kappa_2.$$

Due to the Robbins-Monro condition, the right term converges to real number. Since $\sum_t \alpha_t \eta = \infty$ in the left term, the following must be satisfied:

$$\sum_{t \in \mathcal{N}} \left( \alpha_t \left( \sum_i \nu_{t,i}^a (J_{R_i}(\pi_f) - J_{R_i}(\theta_t)) - \eta \right) \right) = -\infty,$$

which results in $\sum_{t \in \mathcal{N}} \alpha_t = \infty$. Now, let us define a preference $\omega$, where $\omega_i := \bar{\nu}_i^a$. By replacing $\mu$ in (20) with $\pi_\omega^*$,

$$\sum_{t \in \mathcal{N}} \left( \alpha_t \sum_i \nu_{t,i}^a (J_{R_i}(\pi_\omega^*) - J_{R_i}(\theta_t)) \right) - \sum_{t \notin \mathcal{N}} \left( \alpha_t \sum_k \lambda_{t,k}^b (J_{C_k}(\pi_\omega^*) - J_{C_k}(\theta_t)) \right)$$

$$\leq D_{\mathrm{KL}}(\pi_\omega^* \| \pi_0) + \kappa_2 \sum_t \alpha_t^2.$$

$$\Rightarrow \sum_{t \in \mathcal{N}} \left( \alpha_t \sum_i \nu_{t,i}^a (J_{R_i}(\pi_\omega^*) - J_{R_i}(\theta_t)) \right) \leq D_{\mathrm{KL}}(\pi_\omega^* \| \pi_0) + \kappa_2 \sum_t \alpha_t^2.$$

Since the right term converges to a real number and $\sum_{t \in \mathcal{N}} \alpha_t = \infty$, the following must be satisfied:

$$\lim_{t \to \infty} \sum_i \nu_{t,i}^a (J_{R_i}(\pi_\omega^*) - J_{R_i}(\theta_{\mathcal{N}_t})) = 0,$$

where $\mathcal{N}_t$ is the $t$th element of $\mathcal{N}$. Consequently,

$$\sum_i \bar{\nu}_i^a J_{R_i}(\pi_\omega^*) = \sum_i \bar{\nu}_i^a \lim_{t \to \infty} J_{R_i}(\pi_{\mathcal{N}_t}),$$

which means that $\lim_{t \to \infty} \pi_{\mathcal{N}_t}$ is also a LS optimal policy for the preference $\omega$. Due to Lemma B.1, the policy converges to a CP optimal policy. □

## B.3 PROOF OF THEOREM 4.3

**Theorem 4.3.** *If $H$ is the Fisher information matrix and a policy $\pi_{\theta_t}$ is updated as $\theta_{t+1} = \theta_t + g_\omega^{\mathrm{ag}}$, where $g_\omega^{\mathrm{ag}}$ is defined in (9), it converges to a CP optimal policy of (1).*

*Proof.* This theorem can be proved by identifying sequences $\nu_{t,i}^a, \nu_{t,i}^b, \lambda_{t,k}^a, \lambda_{t,k}^b$ that satisfy all conditions mentioned in Theorem 4.2. By examining (9), $\nu_{t,i}^a, \nu_{t,i}^b, \lambda_{t,k}^a, \lambda_{t,k}^b$, and $\alpha_t$ can be deduced as follows:

$$\nu_{t,i}^a = \frac{\nu_i^*}{\sum_{j=1}^N \nu_j^*}, \nu_{t,i}^b = 0, \lambda_{t,k}^b = \frac{\lambda_k^*}{\sum_{j=1}^M \lambda_j^*},$$

$$\lambda_{t,k}^a = \min(\max(||\bar{g}_\omega^{\mathrm{ag}}||, g_{\min}), g_{\max}) \cdot \min\left(\frac{\lambda_k^*}{\epsilon_t \sum_{i=1}^N \nu_i^*}, \lambda_{\max}\right),$$

$$\alpha_t = \frac{\epsilon_t}{\min(\max(||\bar{g}_\omega^{\mathrm{ag}}||, g_{\min}), g_{\max})}.$$

Substituting the deduced sequences into (9), we can obtain the same formulation as (8). The deduced sequences satisfy that $\sum \nu_{t,i}^a = 1$, $\sum \lambda_{t,k}^b = 1$, $\nu_{t,i}^b \in \mathbb{R}_{\geq 0}$ is bounded, $0 \leq \lambda_{t,k}^a \leq g_{\max}\lambda_{\max}$ is also bounded, and $\alpha_t \in [\epsilon_t/g_{\max}, \epsilon_t/g_{\min}]$ follows the Robbins-Monro condition. Additionally, since $\lambda_k^*$ is an optimal point of the dual problem, it satisfies the KKT condition, which results in $\lambda_k^* = 0$ if $J_{C_k}(\theta_t) < d_k$. Consequently, $\lambda_{t,k}^b(J_{C_k}(\theta_t) - d_k) \geq 0$. Now, we need to check whether the $\nu_{t,i}^a$ converges to a real number. Given that the learning rate follows the Robbins-Monro condition, the policy converges to a specific policy. Since (5) becomes invariant at this point of convergence, the solution of the dual problem is also fixed. Consequently, $\nu_{t,i}^a$ converge to a specific value. $\square$

## B.4 CONVERGENCE RATE

In order to derive the convergence rate, we adopted the approach presented in the CRPO paper (Xu et al., 2021). To apply this approach, we slightly modified the policy update rule from Theorem 4.2 as follows:

$$\theta_{t+1} = \theta_t + \begin{cases} \alpha F^\dagger(\theta_t)\left(\sum_{i=1}^N \nu_{t,i}^a g_i - \alpha \sum_{k=1}^M \lambda_{t,k}^a b_k\right) & \text{if } J_{C_k}(\theta_t) \leq d_k + \eta \; \forall k, \\ \alpha F^\dagger(\theta_t)\left(\alpha \sum_{i=1}^N \nu_{t,i}^b g_i - \sum_{k=1}^M \lambda_{t,k}^b b_k\right) & \text{otherwise}, \end{cases} \quad (21)$$

where $\eta > 0$ is called a slack variable, and the key differences are that the recovery step is only taken when the constraints are violated by more than the slack variable, and the learning rate $\alpha$ is fixed over time steps. With these slight modifications, we can derive the following convergence rate.

**Theorem B.2.** *Let us set the learning rate $\alpha$ and the slack variable $\eta$ as follows:*

$$\alpha = 1/\sqrt{T}, \; \eta = 2(D_{\mathrm{KL}}(\pi_\omega^*||\pi_0) + \kappa_2)/\sqrt{T},$$
$$\text{where } \omega = (\bar{\nu}_1^a, ..., \bar{\nu}_N^a), \; \kappa_2 := \kappa_1 + 2R_{\max}\lambda_{\max}\max(N, M)/(1 - \gamma),$$
$$\kappa_1 := 2R_{\max}^2 \max(\alpha, 1)^2 (N + M)^2 \lambda_{\max}^2 \sqrt{|S||A|}/(1 - \gamma)^3.$$

*Then, the followings are satisfied:*

$$\mathbb{E}_{t \sim \mathcal{N}}\left[\sum_i \nu_{t,i}^a \left(J_{R_i}(\pi_\omega^*) - J_{R_i}(\theta_t)\right)\right] \leq 2\frac{D_{\mathrm{KL}}(\pi_\omega^*||\pi_0) + \kappa_2}{\sqrt{T}},$$

$$\mathbb{E}_{t \sim \mathcal{N}}[J_{C_k}(\theta_t)] - d_k \leq 2\frac{D_{\mathrm{KL}}(\pi_\omega^*||\pi_0) + \kappa_2}{\sqrt{T}} \; \forall k.$$

*Proof.* By substituting $|J_{R_i}(\pi_\omega^*) - J_{R_i}(\theta_t)|, |J_{C_k}(\pi_\omega^*) - J_{C_k}(\theta_t)| \leq 2R_{\max}/(1 - \gamma)$ into (20),

$$\sum_{t \in \mathcal{N}}\left(\alpha \sum_i \nu_{t,i}^a (J_{R_i}(\pi_\omega^*) - J_{R_i}(\theta_t)) - 2\alpha^2 M \lambda_{\max} R_{\max}/(1 - \gamma)\right)$$
$$+ \sum_{t \notin \mathcal{N}} \left(-2\alpha^2 N \lambda_{\max} R_{\max}/(1 - \gamma) + \alpha\eta\right) \leq D_{\mathrm{KL}}(\pi_\omega^*||\pi_0) + \kappa_1 \alpha^2 T.$$

$$\Rightarrow \alpha \sum_{t \in \mathcal{N}} \sum_i \nu_{t,i}^a (J_{R_i}(\pi_\omega^*) - J_{R_i}(\theta_t)) + \sum_{t \notin \mathcal{N}} \alpha \eta \le D_{\mathrm{KL}}(\pi_\omega^* || \pi_0) + \kappa_2 T \alpha^2. \tag{22}$$

By using the definitions of $\alpha$ and $\eta$, the following is satisfied:

$$\alpha \eta T / 2 \ge D_{\mathrm{KL}}(\pi_\omega^* || \pi_0) + \kappa_2 \alpha^2 T.$$

If we assume that $\mathcal{N} = \emptyset$, from (22), we have:

$$\alpha \eta T \le D_{\mathrm{KL}}(\pi_\omega^* || \pi_0) + \kappa_2 \alpha^2 T \le \alpha \eta T / 2,$$

which leads to a contradiction. As a result, $\mathcal{N} \ne \emptyset$. Additionally, if we assume that $|\mathcal{N}| < T/2$ and $\sum_{t \in \mathcal{N}} \sum_i \nu_{t,i}^a (J_{R_i}(\pi_\omega^*) - J_{R_i}(\theta_t)) > 0$, we have:

$$\alpha \eta T / 2 < \sum_{t \notin \mathcal{N}} \alpha \eta < \alpha \sum_{t \in \mathcal{N}} \sum_i \nu_{t,i}^a (J_{R_i}(\pi_\omega^*) - J_{R_i}(\theta_t)) + \sum_{t \notin \mathcal{N}} \alpha \eta \le \alpha \eta T / 2,$$

which leads to a contradiction. As a result, one of the followings are satisfied: **1)** $|\mathcal{N}| \ge T/2$ or **2)** $\sum_{t \in \mathcal{N}} \sum_i \nu_{t,i}^a (J_{R_i}(\pi_\omega^*) - J_{R_i}(\theta_t)) \le 0$. If $|\mathcal{N}| \ge T/2$,

$$\mathbb{E}_{t \sim \mathcal{N}} \left[ \sum_i \nu_{t,i}^a (J_{R_i}(\pi_\omega^*) - J_{R_i}(\theta_t)) \right] = \frac{\sum_{t \in \mathcal{N}} \sum_i \nu_{t,i}^a (J_{R_i}(\pi_\omega^*) - J_{R_i}(\theta_t))}{|\mathcal{N}|}$$

$$\le 2 \frac{D_{\mathrm{KL}}(\pi_\omega^* || \pi_0)/\alpha + \kappa_2 T \alpha}{T} = 2 \frac{D_{\mathrm{KL}}(\pi_\omega^* || \pi_0) + \kappa_2}{\sqrt{T}}.$$

If $\sum_{t \in \mathcal{N}} \sum_i \nu_{t,i}^a (J_{R_i}(\pi_\omega^*) - J_{R_i}(\theta_t)) \le 0$, the above inequality also holds. As a result, the performance gap between the LS optimal policy $\pi_\omega^*$ and the current policy $\pi_t$ is bounded by $\mathcal{O}(1/\sqrt{T})$. For constraint satisfaction, we derive the following bound:

$$\mathbb{E}_{t \sim \mathcal{N}}[J_{C_i}(\theta_t)] - d_i \le \eta = 2 \frac{D_{\mathrm{KL}}(\pi_\omega^* || \pi_0) + \kappa_2}{\sqrt{T}},$$

which implies that the constraint violation is also bounded by $\mathcal{O}(1/\sqrt{T})$. $\qquad \square$

## C  EXPERIMENTAL DETAILS

In this section, we describe the details of each task, including Safety Gymnasium (Ji et al., 2023), legged robot locomotion (Kim et al., 2023), and Multi-Objective Gymnasium (Felten et al., 2023). We then provide a detailed explanation of the performance metrics named hypervolume and sparsity. Finally, we present implementation details, such as network architectures and hyperparameter settings.

### C.1  TASK DETAILS

#### C.1.1  SAFETY GYMNASIUM

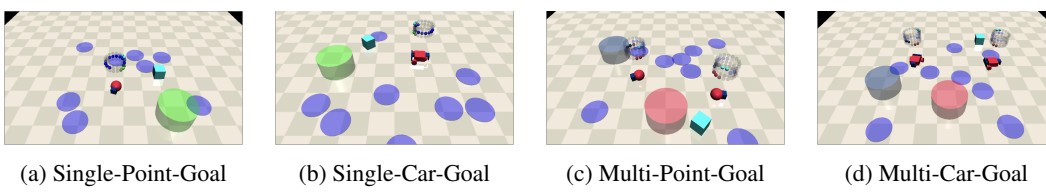

|     (a) Single-Point-Goal     |     (b) Single-Car-Goal     |     (c) Multi-Point-Goal     |     (d) Multi-Car-Goal     |

Figure 11: **Snapshots of the Safety-Gymnasium tasks.** There are two types of robots: point and car. These tasks aim to control the robots to reach goals while avoiding hazardous areas, colored purple. In the Single-Point-Goal and Single-Car-Goal tasks, a single goal, shown in green, is randomly spawned. Conversely, the Multi-Point-Goal and Multi-Car-Goal tasks have two goals, shown in blue and red, and if an agent reaches any of them, a reward is given to the agent.

In the Safety Gymnasium (Ji et al., 2023), we utilize the single-agent safe navigation tasks: `SafetyPointGoal1-v0` and `SafetyCarGoal1-v0`, and the multi-agent safe navigation

tasks: `SafetyPointMultiGoal1-v0` and `SafetyCarMultiGoal1-v0`. Snapshots of each task are shown in Figure 11.

In the single-agent tasks, the observation space includes velocity, acceleration, and Lidar of the goal and hazards, and the dimensions are 60 and 72 for the point and car robots, respectively. The action space for both robots is two-dimensional. There are eight hazard areas and one goal, and hazard areas are randomly spawned at the beginning of each episode, while goals are randomly placed whenever the agent reaches the current goal. These tasks originally had a single objective: maximizing the number of goals reached. However, in order to have multiple objectives, we have modified them to include a second objective: minimizing energy consumption. Therefore, the reward is two-dimensional, and we use the original definition for the first reward (please refer to Ji et al. (2023)). The second reward is defined as follows:

$$R_2(s, a, s') := -\frac{1}{|A|} \sum_{i=1}^{|A|} (a_i/10)^2. \tag{23}$$

Also, we use the original definition of the cost function, which gives one if the agent enters hazardous areas and zero otherwise.

The multi-agent tasks have two agents and two goals, and we need to control both agents to reach goals as much as possible while avoiding hazardous areas. The original implementation uses a dictionary type for observations and actions, so we have modified them to be an array type. Additionally, in the original setup, each goal is pre-assigned to a specific agent, reducing the difficulty of solving multi-agent tasks. Therefore, we have modified the implementation so that each goal is not pre-assigned, allowing each agent to compete to achieve goals. The observation space contains the same information from the single-agent tasks for the two agents, as well as additional information: Lidar of the first and second goals. The dimensions of observation space are then 152 and 176 for the point and car robots, respectively. The action space is four-dimensional, which is doubled from the single-agent tasks to control both agents. These tasks have two objectives: maximizing the number of goals reached by each agent. Also, there are two constraints: avoiding hazardous areas for each agent. Since there are no additional objectives or constraints, the original definitions of the reward and cost functions are used without modification.

### C.1.2 LEGGED ROBOT LOCOMOTION

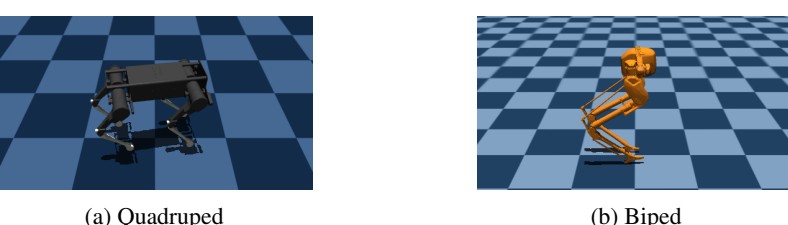

(a) Quadruped      (b) Biped

Figure 12: **Snapshots of the legged robot locomotion tasks.** Robots aim to follow a given command while ensuring they do not fall over.

The legged robot locomotion tasks (Kim et al., 2023) aim to control bipedal or quadrupedal robots so that their velocity matches a randomly sampled command. This command specifies the target linear velocities in the x and y-axis directions and the target angular velocity in the z-axis direction, denoted by $(v_x^{\text{cmd}}, v_y^{\text{cmd}}, \omega_z^{\text{cmd}})$. Snapshots of these tasks are shown in Figure 12. The quadrupedal robot has 12 joints and 12 motors, and the bipedal robot has 14 joints and 10 motors. Each robot is operated by a PD controller that follows the target position of the motors, and the target position is given as an action. Hence, the number of motors corresponds to the dimension of action space. In order to provide enough information for stable control, observations include the command, linear and angular velocities of the robot base, and the position and velocity of each joint. As a result, the dimensions of the observation space are 160 for the quadruped and 132 for the biped. Originally, these tasks have a single objective of following a given command and three constraints: 1) maintaining body balance, 2) keeping the height of CoM (center of mass) above a certain level, and 3) adhering to pre-defined foot contact timing. Therefore, we use the original implementation for the observation, action, and cost functions but modify the reward function to have multiple objectives. The modified version has

two objectives: 1) reducing the difference between the current velocity and the command and 2) minimizing energy consumption. Then, the reward function is defined as follows:

$$R_1(s, a, s') := 1 - (v_x^{\text{base}} - v_x^{\text{cmd}})^2 - (v_y^{\text{base}} - v_y^{\text{cmd}})^2 - (\omega_z^{\text{base}} - \omega_z^{\text{cmd}})^2,$$

$$R_2(s, a, s') := 1 - \frac{1}{J} \sum_{j=1}^{J} \left( \frac{\tau_j}{M_{\text{robot}}} \right)^2, \tag{24}$$

where $J$ is the number of joints, $\tau_j$ is the torque applied to the $j$th joint, and $M_{\text{robot}}$ is the mass of the robot. However, in the simulation of the bipedal robot, obtaining valid torque information is difficult due to the presence of closed loops in the joint configuration. To address this issue, we modify the second reward for the bipedal robot to penalize the action instead of the joint torque as follows:

$$R_2(s, a, s') := 1 - \frac{1}{|A|} \sum_{i=1}^{|A|} a_i^2. \tag{25}$$

### C.1.3 MULTI-OBJECTIVE GYMNASIUM

Table 1: Specifications of MO-Gymnasium tasks.

|  | Observation Space | Action Space | # of Objectives (Entries) |
|---|---|---|---|
| Half-Cheetah | $\subseteq \mathbb{R}^{17}$ | $\subseteq \mathbb{R}^{6}$ | 2 (velocity, energy) |
| Hopper | $\subseteq \mathbb{R}^{11}$ | $\subseteq \mathbb{R}^{3}$ | 3 (velocity, height, energy) |
| Humanoid | $\subseteq \mathbb{R}^{376}$ | $\subseteq \mathbb{R}^{17}$ | 2 (velocity, energy) |
| Ant | $\subseteq \mathbb{R}^{27}$ | $\subseteq \mathbb{R}^{8}$ | 3 ($x$-velocity, $y$-velocity, energy) |
| Walker2d | $\subseteq \mathbb{R}^{17}$ | $\subseteq \mathbb{R}^{6}$ | 2 (velocity, energy) |
| Swimmer | $\subseteq \mathbb{R}^{8}$ | $\subseteq \mathbb{R}^{2}$ | 2 (velocity, energy) |

We utilize the Multi-Objective (MO) Gymnasium (Felten et al., 2023) to evaluate the proposed method in MORL tasks which have no constraints. Among several tasks, we use the MuJoCo tasks with continuous action space, and there are six tasks available: `mo-hopper-v4`, `mo-humanoid-v4`, `mo-halfcheetah-v4`, `mo-ant-v4`, `mo-walker2d-v4`, and `mo-swimmer-v4`. Description of the observation space, action space, and the number of objectives for these tasks are summarized in Table 1, and for more details, please refer to Felten et al. (2023).

### C.2 METRIC DETAILS

In this section, we explain in detail the performance metrics used in the main text: hypervolume (HV) and normalized sparsity ($\overline{\text{SP}}$). First, we need to prepare an estimated CP front to calculate such metrics. To do this, we pre-sample a fixed number of equally spaced preferences and roll out the trained universal policy for each pre-sampled preference to estimate objective and constraint values. If the constraint values do not exceed the thresholds, we store the objective values, $(J_{R_1}(\theta), ..., J_{R_N}(\theta))$, in a set. Once this process is completed for all preferences, we can construct an estimated CP front by extracting elements from the collected set that are not dominated by any others. The objective values are then normalized using the minimum and maximum values from the entire dataset, effectively minimizing the impact of scale of each objective. Now, we introduce details on each metric.

HV is the volume of the area surrounding a given reference point, $r$, and an estimated CP front, $P$, as defined in (12) and is visualized in Figure 4(a). However, as seen in Figure 4(a) or (12), the metric value varies depending on what reference point is used. Therefore, we do not set the reference point arbitrarily but use a method of determining the reference point from the results of all algorithms, as shown in Figure 4(b). For a detailed explanation, suppose that we have a set of estimated CP fronts, $\{P_i\}_{i=1}^{K}$, where $P_i$ is the estimated front obtained from the $i$th algorithm. Then, we obtain the union of $\{P_i\}_{i=1}^{K}$, denoted by $P'$, and extract elements that are not dominated by any others in $P'$ to find the Pareto front of the entire set, $\text{PF}(P')$, whose elements are circled in red in Figure 4(b). Finally,

we get the reference point whose $i$th value is $\min(\{p_i|\forall p \in \mathrm{PF}(P')\})$. Through this process, the reference point can be automatically set after obtaining results from all algorithms for each task. In addition, the HV value can be zero for some algorithms whose elements are entirely dominated by $\mathrm{PF}(P')$. An example of this is Algorithm 3 in Figure 4(b).

As mentioned in the main text, sparsity (SP) (Xu et al., 2020) has a correlation with HV is defined as follows:

$$\mathrm{SP}(P) := \frac{1}{|P|-1} \sum_{j=1}^{N} \sum_{i=1}^{|P|-1} (\tilde{P}_j[i] - \tilde{P}_j[i+1])^2, \tag{26}$$

where $\tilde{P}_j := \mathrm{Sort}(\{p[j]|\forall p \in P\})$. An increase in HV implies that the elements of the CP front are moving further away from each other, which increases SP. To remove this correlation, we provide a normalized version as defined in (13), and the calculation process is illustrated in Figure 4(c). Given an estimated CP front, $P$, we normalize each estimated objective value so that the minimum value corresponds to zero and the maximum to one. The normalized objectives are denoted as $\overline{J_{R_i}}(\theta)$. We then calculate the average of the squares of the distances between elements in this normalized space. This normalized version still produces a large value when a CP front is densely clustered in particular areas while being sparsely distributed overall. Therefore, it maintains the ability to measure sparsity while removing the correlation with HV.

## C.3 IMPLEMENTATION DETAILS

In this section, several implementation details including hyperparameter settings and network structures are provided. Before that, we report information on the computational resources. In all experiments, we used a PC equipped with an Intel Xeon CPU E5-2680 and an NVIDIA TITAN Xp GPU. The average training time in the single point goal task is presented in Table 2.

Table 2: Training time on the point single goal task.

|  | CAPQL | PD-MORL | LP3 | CoMOGA (ours) |
|---|---|---|---|---|
| Time | 16h 49m 29s | 16h 40m 9s | 24h 4m 34s | 17h 27m 58s |

### C.3.1 IMPLEMENTATION DETAILS OF CONSTRAINED OPTIMIZATION PROBLEM

CoMOGA aggregates the gradients of the objective and constraint functions by solving the following quadratic programming (QP) problem:

$$\bar{g}_\omega^{\mathrm{ag}} = \mathrm{argmin}_{\Delta\theta} \Delta\theta^T H \Delta\theta \ \mathbf{s.t.} \ \omega_i e_i \le g_i^T \Delta\theta, \ b_k^T \Delta\theta + J_{C_k}(\theta_{\mathrm{old}}) \le d_k.$$

As we mentioned in Section 4.4, the QP problem holds strong duality, so the solution can be expressed as follows: $\bar{g}_\omega^{\mathrm{ag}} = H^{-1}(\sum_{i=1}^{N} \nu_i^* g_i - \sum_{k=1}^{M} \lambda_k^* b_k)$, where $\nu_i^*$ and $\lambda_k^*$ are the optimal Lagrange multipliers of the following dual problem:

$$\begin{aligned}
\nu^*, \lambda^* = \mathop{\mathrm{argmax}}_{\nu \ge 0, \lambda \ge 0} &-\frac{1}{2} \left( \sum_i \nu_i g_i - \sum_k \lambda_k b_k \right)^T H^{-1} \left( \sum_i \nu_i g_i - \sum_k \lambda_k b_k \right) \\
&+ \sum_i \nu_i \omega_i e_i + \sum_k \lambda_k (J_{C_k}(\theta_{\mathrm{old}}) - d_k).
\end{aligned} \tag{27}$$

In order to solve this problem (27), we first compute $g_i$ and $b_k$, which can be obtained by backpropagation using (11). Next, we need to calculate $H^{-1}g_i$. If $H$ is an identity matrix, $H^{-1}g_i$ simplifies to $g_i$ itself, and no additional computation is required. When $H$ is the Fisher Information Matrix (FIM), this calculation aligns with the policy gradient method used in the TRPO algorithm (Schulman et al., 2015). Consequently, we employ the conjugate gradient (CG) method, as in TRPO, to approximate $H^{-1}g_i$. The CG method iteratively approximates $H^{-1}g_i$ within a finite number of steps, and in our implementation, we used 10 iterations. For further details about the implementation, we refer to the TRPO code in the Stable-Baselines3 repository (Raffin et al., 2021).

For simplicity of notation, let $g_{k+N} = -b_k$ for $k = 1, ..., M$. Once $g_i$ and $H^{-1}g_i$ are computed, we construct a $(N + M) \times (N + M)$ matrix $P = \{p_{ij}\}$, where $p_{ij} = g_i^T H^{-1} g_j$. We also compute

a $(N + M)$-dimensional vector $q$, where $q_i = -\omega_i e_i$, if $i \leq N$, $d_k - J_{C_k}(\theta_{\text{old}})$, otherwise. Then, (27) can be rewritten as:

$$\nu_i^* = x_i^* \ \forall i, \ \lambda_k^* = x_{N+k}^* \ \forall k, \ \text{where} \ x^* = \arg\min_{x \geq 0} \frac{1}{2} x^T P x + q^T x. \tag{28}$$

Since this problem is formulated in a low-dimensional space with $N + M$ variables, it can be efficiently solved using a QP solver (Caron et al., 2024) employed in numerical optimization. After solving the optimal dual variables, the aggregated gradient can be calculated as: $\bar{g}_\omega^{\text{ag}} = \sum_{i=1}^{N} \nu_i^* H^{-1} g_i - \sum_{k=1}^{M} \lambda_k^* H^{-1} b_k$.

Regarding the complexity of solving the constrained optimization problem (COP), the QP solver is highly efficient and well-established, allowing it to solve most problems within milliseconds. Thus, the time required for the QP solver can be assumed to be constant. The dominant factor in computational complexity is therefore determined by the process of setting up (28). This involves calculating the inner products between gradients, which are performed $(N + M)^2$ times. Consequently, the complexity is proportional to $(N + M)^2 P$ where $P$ is the number of policy parameters. Additionally, the Conjugate Gradient (CG) method involves a finite number of iterations. As such, its computational cost can be treated as a constant relative to the gradient calculations. In summary, the overall complexity of solving the COP is $\mathcal{O}((N + M)^2 P)$.

### C.3.2   MORL TO CMORL USING THE LAGRANGIAN METHOD

In order to apply linear scalarization-based MORL methods to CMORL, it is required to solve the following constrained optimization problem given a preference $\omega$:

$$\max_\theta \sum_{i=1}^{N} \omega_i J_{R_i}(\theta) \ \text{s.t.} \ J_{C_k}(\theta) \leq d_k \ \forall k \in \{1, .., M\}. \tag{29}$$

As explained in the main text, the constraints of the above problem can be handled by converting it into a Lagrange dual problem, and the dual problem is written as follows:

$$\max_{\lambda \geq 0} \min_\theta - \sum_{i=1}^{N} \omega_i J_{R_i}(\theta) + \sum_{k=1}^{M} \lambda_k \cdot (J_{C_k}(\theta) - d_k), \tag{30}$$

where $\lambda_k$ are Lagrange multipliers. The multiplies should be learned separately for each preference, but preferences have continuous values. Therefore, we instead parameterize the multiplier using neural networks as $\lambda_\phi(\omega)$, where $\phi$ is a parameter. Then, the above problem can be solved by concurrently updating the policy parameter and the Lagrange multipliers using the following loss functions:

$$\min_\theta L(\theta) := - \sum_{i=1}^{N} \omega_i J_{R_i}(\theta) + \sum_{k=1}^{M} \lambda_k J_{C_k}(\theta),$$

$$\min_\phi L(\phi) := - \sum_{k=1}^{M} \lambda_{k,\phi}(\omega)(J_{C_k}(\theta) - d_k) \ \text{s.t.} \ \lambda_{k,\phi}(\omega) \geq 0, \tag{31}$$

where the multipliers can be forced to non-negative values using some activation functions, such as softplus and exponential functions. The policy loss in (31) is constructed by adding $\sum \lambda_k J_{C_k}(\theta)$ to the loss from the unconstrained MORL problem. Therefore, the implementation of the existing MORL algorithms can be easily extended to a CMORL algorithm by adding $\sum \lambda_k J_{C_k}(\theta)$ to the original policy loss.

Additionally, the PD-MORL algorithm (Basaklar et al., 2023) uses a preference alignment technique. However, for a fair comparison, we did not employ this technique as it necessitates 1) pre-training to identify the objective direction corresponding to key preferences and 2) additional evaluation steps during the main training process to adapt the objective direction. Nevertheless, to enrich our discussion, we provide experimental results of the PD-MORL algorithm with preference alignment in Appendix D.

### C.3.3   NETWORK ARCHITECTURE

The proposed method and baselines are based on the actor-critic framework, requiring policy and critic networks. Additionally, since the baselines use the Lagrangian method to handle the constraints, they use multiplier networks, as mentioned in Appendix C.3.2. We have implemented these networks as fully connected networks (FCNs), and their structures are presented in Table 3. In

Table 3: Network structures.

|  | Parameter | Value |
|---|---|---|
| Policy network | Hidden layer
Activation
Normalization | (512, 512)
LeakyReLU
✗ |
| Critic network | Hidden layer
Activation
Normalization | (512, 512)
LeakyReLU
✗ |
| Quantile distributional
critic network | Hidden layer
Activation
Normalization
# of quantiles | (512, 512)
LeakyReLU
LayerNorm
25 |
| Multiplier network | Hidden layer
Activation
Normalization | (512,)
LeakyReLU
✗ |

the unconstrained tasks, MO-Gymnasium, we use standard critic networks that output scalar values for given observations and preferences. However, the standard critic networks usually estimate the objective or constraint functions with large biases, making it challenging to satisfy the constraints. Therefore, we use quantile distributional critics (Dabney et al., 2018) to lower the estimation biases in the Safety-Gymnasium and legged robot locomotion experiments. The quantile distribution critic outputs several quantile values of the discounted sum of the reward or cost functions for given observations and preferences. Then, the objective or constraint functions can be estimated by averaging the outputted quantiles. For the proposed method and LP3 that have similar frameworks to TRPO (Schulman et al., 2015), TD($\lambda$) method (Kim et al., 2023) is used to train the distributional critic networks. For PD-MORL and CAPQL, the distributional critic networks are trained by reducing the Wasserstein distance between the current quantiles and the truncated one-step TD targets, as in TQC (Kuznetsov et al., 2020). The policy network, similar to other RL algorithms dealing with continuous action spaces (Schulman et al., 2015; Haarnoja et al., 2018; Fujimoto et al., 2018), outputs the mean and diagonal variance of a normal distribution. However, in the locomotion tasks, the policy networks are modified to output only the mean value by fixing the diagonal variance in order to lower the training difficulty.

### C.3.4   HYPERPARAMETER SETTINGS

We report the hyperparameter settings for the CMORL tasks (Safety-Gymnasium, Locomotion) in Table 4 and the settings for the MORL tasks (MO-Gymnasium) in Table 5.

Table 4: Hyperparameters for CMORL tasks.

| | CAPQL | PD-MORL | LP3 | **CoMOGA (Ours)** |
|---|---|---|---|---|
| Discount factor | 0.99 | 0.99 | 0.99 | 0.99 |
| Length of replay buffer | 1000000 | 1000000 | 100000 | 100000 |
| Steps per update | 10 | 10 | 1000 | 1000 |
| Batch size | 256 | 256 | 10000 | 10000 |
| Optimizer | Adam | Adam | Adam | Adam |
| Policy learning rate | $3 \times 10^{-4}$ | $3 \times 10^{-4}$ | - | $3 \times 10^{-4}$ |
| Critic learning rate | $3 \times 10^{-4}$ | $3 \times 10^{-4}$ | $3 \times 10^{-4}$ | $3 \times 10^{-4}$ |
| Multiplier learning rate | $1 \times 10^{-5}$ | $1 \times 10^{-5}$ | $3 \times 10^{-4}$ | - |
| Soft update ratio | 0.005 | 0.005 | - | - |
| # of quantiles to truncate | 2 | 2 | - | - |
| # of HER samples | - | 3 | - | - |
| Angle loss coefficient | - | 10 | - | - |
| Explore and target action noise scale | - | (0.1, 0.2) | - | - |
| # of action samples | - | - | 20 | - |
| TD($\lambda$) factor | - | - | 0.97 | 0.97 |
| # of target quantiles | - | - | 50 | 50 |
| # of preference samples | - | - | 10 | 10 |
| Local region size | - | - | 0.05 | 0.05 |
| $g_{\min}, g_{\max}$ | - | - | - | $(0, \infty)$ |
| $\lambda_{\max}$ | - | - | - | $\infty$ |
| $H$ matrix | - | - | Fisher information | Fisher information |

Table 5: Hyperparameters for MORL tasks.

| | CAPQL | PD-MORL | LP3 | **CoMOGA (Ours)** |
|---|---|---|---|---|
| Discount factor | 0.99 | 0.99 | 0.99 | 0.99 |
| Length of replay buffer | 1000000 | 1000000 | 100000 | 1000000 |
| Steps per update | 1 | 1 | 1000 | 10 |
| Batch size | 256 | 256 | 10000 | 256 |
| Optimizer | Adam | Adam | Adam | Adam |
| Policy learning rate | $3 \times 10^{-4}$ | $3 \times 10^{-4}$ | - | $3 \times 10^{-4}$ |
| Critic learning rate | $3 \times 10^{-4}$ | $3 \times 10^{-4}$ | $3 \times 10^{-4}$ | $3 \times 10^{-4}$ |
| Soft update ratio | 0.005 | 0.005 | - | 0.005 |
| # of HER samples | - | 3 | - | - |
| Angle loss coefficient | - | 10 | - | - |
| Explore and target action noise scale | - | (0.1, 0.2) | - | - |
| # of action samples | - | - | 20 | - |
| TD($\lambda$) factor | - | - | 0.97 | - |
| # of preference samples | - | - | 10 | 10 |
| Local region size | - | - | 0.14 | 0.05 |
| $g_{\min}, g_{\max}$ | - | - | - | $(0, \infty)$ |
| $\lambda_{\max}$ | - | - | - | $\infty$ |
| $H$ matrix | - | - | Fisher information | Identity matrix |

# D    ADDITIONAL EXPERIMENTAL RESULTS

In this section, we present additional experimental results for CAGrad (Liu et al., 2021) and PD-MORL with preference alignment (Basaklar et al., 2023), which is mentioned in Appendix C.3.2 and denoted by *PD-MORL-Norm*, on the MO Gymnasium tasks. The training curves are shown in Figure 13. Additionally, Figure 14 presents the Hypervolume results without normalization. The estimated CP fronts for tasks with two objectives are visualized in Figure 15, where the hopper and ant tasks are excluded as they have three objectives. As mentioned in the appendix C.2, the estimated CP front can be obtained by rolling out a trained universal policy with different preferences, and for the MO Gymnasium tasks, 20 equally spaced preferences are used. Figure 15 show that CAGrad performs similarly to our method, CoMOGA, on the overall tasks. However, as seen in Figure 8, it is evident that CAGrad is significantly outperformed by our proposed method. This observation confirms that extending existing conflict-averse MORL algorithms to CMORL is non-trivial. Additionally, PD-MORL-Norm performs better than PD-MORL in the ant and hopper tasks but worse in the humanoid and walker tasks.

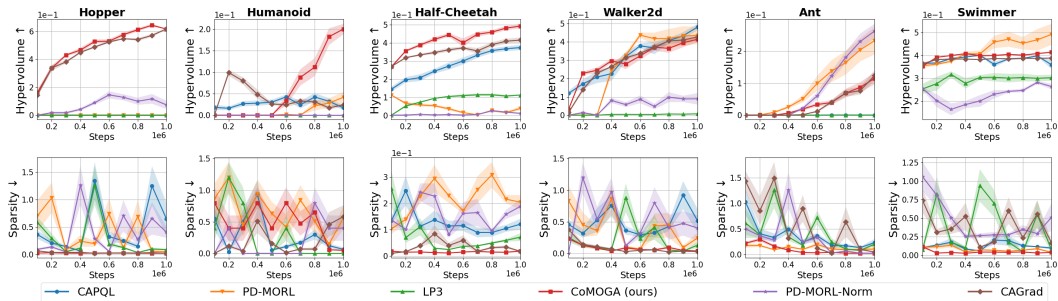

Figure 13: Additional evaluation results of the MO Gymnasium tasks. Each column shows the results of the task corresponding to its title. All algorithms are evaluated at every $10^5$ steps. The bold lines and shaded areas represent the mean and quarter-scaled standard deviation of results from five random seeds, respectively.

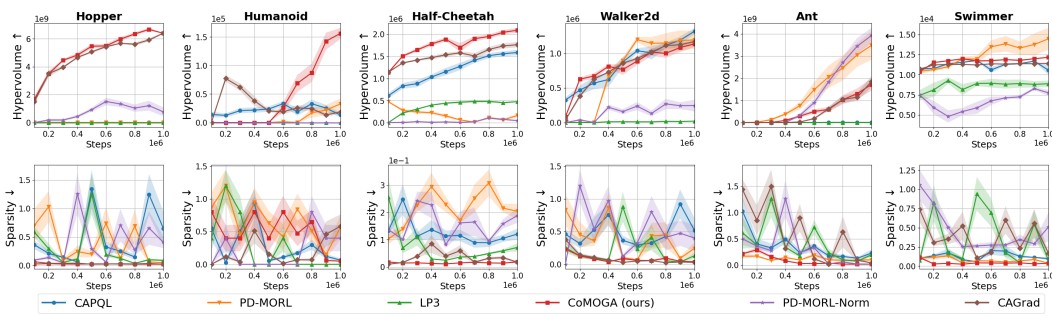

Figure 14: The same results as Figure 13, but with Hypervolume calculated without normalization.

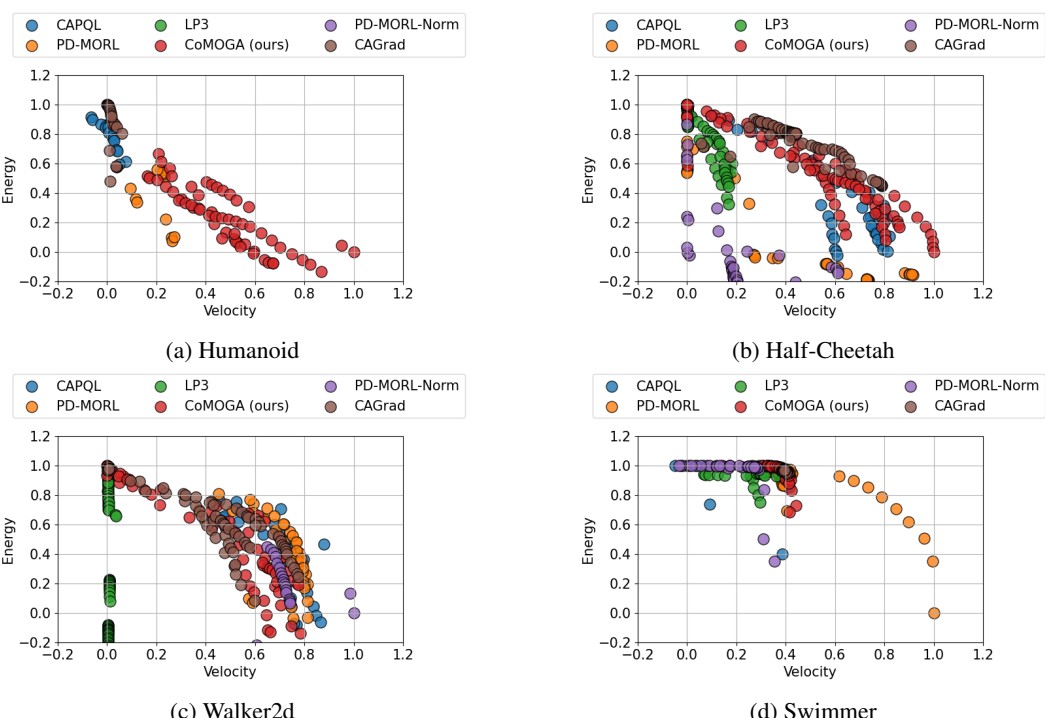

Figure 15: Visualization of estimated CP fronts for MO Gymnasium tasks.

### D.1 Sensitivity Analysis of Hyperparameters

Our method introduces two additional hyperparameters: the local region size $\epsilon$ and the number of preference samples $P$, as specified in Algorithm 1. To provide insights into the sensitivity of the proposed method to these hyperparameters, we conducted experiments on the quadrupedal locomotion task using various values. The results are presented in Figure 16.

For the local region size $\epsilon$, experiments were conducted with values of $0.01$, $0.02$, $0.05$, and $0.1$. The results show that increasing the size accelerates the convergence of HV performance. However, as the size increases, the constraint curves fluctuate, suggesting that handling constraints might become less stable. Despite this, CoMOGA achieves optimal performance and satisfies the constraints in all cases, demonstrating robustness to variations in the local region size.

For the number of preference samples, experiments were conducted with values of $P = 1$, $5$, $10$, and $20$. The performance was lowest at $P = 1$, while the results for $P = 5$, $10$, and $20$ were consistently satisfactory. However, increasing $P$ does not always lead to better performance. This can be attributed to the fact that a larger $P$ reduces the trajectory size corresponding to each preference, potentially limiting effectiveness.

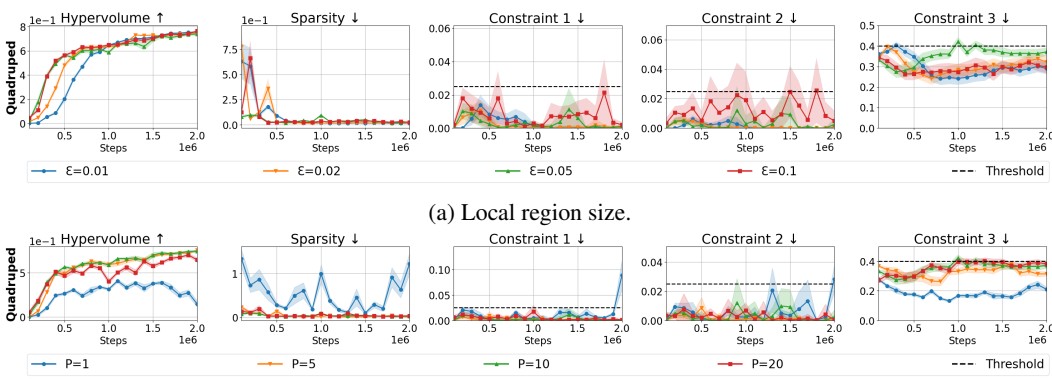

(a) Local region size.

(b) The number of preference samples.

Figure 16: Sensitivity experiments on the quadrupedal locomotion task.

### D.2 Computational Complexity Analysis

In this section, we experimentally analyze the GPU memory usage and computational complexity of solving the constrained optimization problem (COP) in the proposed CoMOGA algorithm. We use Single-Point-Goal, Multi-Point-Goal, and Quadruped tasks, with respective $N + M$ values of three, four, and five. The results are summarized in Table 6. The first two rows of the table report the GPU memory usage dedicated to solving the COP, while the remaining rows present the computation times. Specifically, solving a COP involves the following steps: **1)** calculating the gradients of the objective and constraint functions, **2)** performing gradient operations related to the Hessian matrix, and **3)** solving the COP. We report the time consumed for each of these steps. The results indicate that COP-related computations contribute minimally to both GPU memory usage and computation time. However, Hessian-related operations account for approximately $10\%$ of the total computation time. This suggests that future research could benefit from adopting an approximation approach similar to PPO (Schulman et al., 2017), which avoids explicit Hessian computations, to achieve faster training time.

### D.3 Preference Sampling Strategies

In this section, we provide a detailed explanation of the implementation of $\omega \sim \Omega$ as specified in Algorithm 1. The most straightforward approach for sampling preferences is to draw them from an $N$-dimensional Dirichlet distribution. Another approach is to sample one-hot vectors to represent strong preferences for a specific objective. In this case, the dimension with a value of $1$ can be selected uniformly at random. Lastly, a hybrid strategy can be employed, where preferences are

Table 6: Comparison of GPU memory usage and computational time across tasks.

|  | Single-Point-Goal ($N + M = 3$) | Multi-Point-Goal ($N + M = 4$) | Quadruped ($N + M = 5$) |
|---|---|---|---|
| COP GPU Memory (MB) | 20 (0.91 %) | 40 (1.71 %) | 40 (1.32 %) |
| Total GPU Memory (MB) | 2189 (100 %) | 2333 (100 %) | 3037 (100 %) |
| Gradient Computation (sec) | 0.097 (1.11 %) | 0.156 (1.40 %) | 0.192 (1.94 %) |
| Hessian Computation (sec) | 1.271 (14.50 %) | 1.975 (17.67 %) | 1.324 (13.41 %) |
| QP Solving Time (sec) | 0.012 (0.14 %) | 0.014 (0.13 %) | 0.014 (0.14 %) |
| Training Time per Update (sec) | 8.763 (100 %) | 11.174 (100 %) | 9.870 (100 %) |

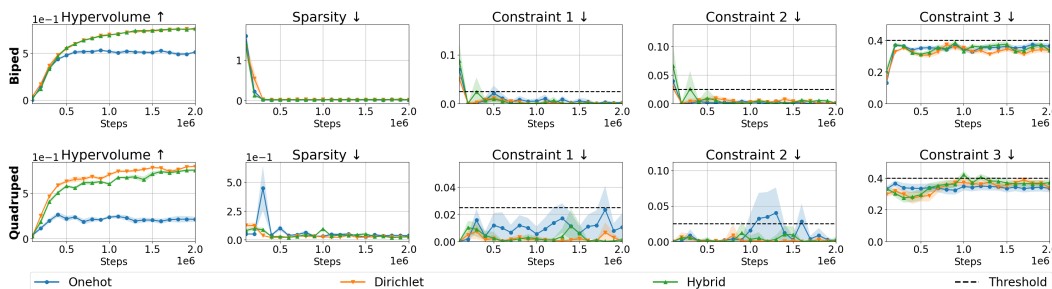

Figure 17: Training results of different preference sampling approaches on the locomotion tasks.

sampled from the Dirichlet distribution with a $50\%$ probability and as one-hot vectors with the remaining $50\%$.

In our implementation, we adopted the hybrid sampling strategy. To investigate the impact of different sampling methods on performance, we compared the results of training legged locomotion tasks using all three strategies. The results are summarized in Figure 17.

As shown in the figure, sampling preferences from the Dirichlet distribution yields the best performance. The hybrid strategy performs almost as well as the Dirichlet sampling, while the one-hot sampling method suffers from severe performance degradation. These findings suggest that the Dirichlet distribution, which enables uniform sampling across the preference space, is the most effective approach.

# E  COMPARISON WITH CONFLICT-AVERSE MTL ALGORITHMS

To compare CoMOGA with existing conflict-averse multi-task learning (MTL) algorithms, we selected MGDA (Désidéri, 2012), PCGrad (Yu et al., 2020), and CAGrad (Liu et al., 2021). The aggregation of multiple gradients in each method can be formulated as follows and is illustrated as in Figure 18:

$$
\begin{aligned}
\text{(MGDA)} \qquad & \max_d \min_i g_i^T d \ \textbf{s.t.} \ ||d|| \leq 1, \\[1mm]
\text{(PCGrad)} \qquad & d = \sum_i g_i^{\text{PC}}, \ \text{where } g_i^{\text{PC}} = g_i - \frac{g_i^T g_j}{||g_j||_2^2} g_j, \\[1mm]
\text{(CAGrad)} \qquad & \max_d \min_i g_i^T d \ \textbf{s.t.} \ ||d - g_0|| \leq c||g_0||, \\[1mm]
\text{(CoMOGA)} \qquad & \min_d ||d|| \ \textbf{s.t.} \ \omega_i \epsilon ||g_i|| \leq g_i^T d,
\end{aligned}
\tag{32}
$$

where CoMOGA is a special case where no safety constraints are presented, and $H$ is the identity matrix. Below, we detail the differences between these methods and the proposed CoMOGA.

- **MGDA.** This method determines a gradient direction that maximizes the improvement of the objective with the smallest increase. However, MGDA has a limitation in that it does not allow the incorporation of task preferences.

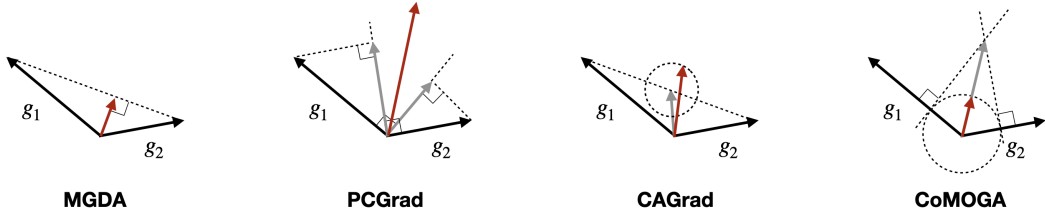

Figure 18: Visualization of gradient aggregation process in conflict-averse MTL algorithms.

- **PCGrad.** This method projects each gradient to remove components in conflict with other gradients before aggregating the projected gradients. However, like MGDA, PCGrad does not support the incorporation of task preferences.

- **CAGrad.** This method can be viewed as an extension of MGDA that enables the inclusion of task preferences by setting $g_0 = \sum_i \omega_i g_i$.

- **CoMOGA.** Finally, the proposed method, CoMOGA, transforms each objective into constraints and solves a quadratic programming (QP) problem to obtain the gradient.

