# OpenReview forum: "Conflict-Averse Gradient Aggregation for Constrained Multi-Objective Reinforcement Learning"
_ICLR.cc/2025/Conference — ICLR 2025 Poster_

### Official Review · Reviewer_6dPa · 2024-10-29

**Soundness:** 3
**Presentation:** 2
**Contribution:** 2
**Rating:** 5
**Confidence:** 3

**Summary:**

This paper introduces a novel multi-objective reinforcement learning (RL) approach that incorporates safety constraints. The authors present a gradient conflict management technique, which formulates multi-objective RL as a constrained optimization problem. This formulation effectively prevents policies from getting trapped in local optima, particularly in tabular settings. Additionally, it enables the seamless integration of safety constraints and facilitates an efficient policy update mechanism.

**Strengths:**

1.***Theoretical Contribution:***

The paper presents an interesting theoretical contribution, providing a comprehensive convergence analysis.

2.***Experimental Coverage:***

The study includes a wide range of experiments within the MORL domain, offering an extensive evaluation of the proposed method.

**Weaknesses:**

1.***Constrained Multi-Objective RL as a Constrained Optimization Problem:***

The authors propose addressing constrained multi-objective reinforcement learning (CMORL) by formulating it as a constrained optimization problem (COP). While COP offers robust theoretical guarantees for convergence, the paper provides limited insights into its practical implementation, which could impact the reproducibility and real-world application of the proposed method.

2.***Inconsistencies in Hypervolume (HV) Scale and Reporting:***
Figure 8 presents hypervolume (HV) results on a significantly smaller scale compared to previous work. For example, in PD-MORL, HV values are reported across training steps up to 10 $e^6$, while in this study, the results are shown only from 1$e^6$ up to 3$e^6$ steps, Which may not reach the convergency.

3.***Different unit to benchmark in HV:***
 With PD-MORL reporting HV values in the unit of 1$e^6$ , whereas this paper uses a unit of 0.1 for HV. These inconsistencies create confusion and lead to the perception that PD-MORL performs worse in this comparison than reported in the original study.

**Questions:**

1. **Limited Practical Implementation Details in Constrained Optimization Problem (COP)**: The authors propose solving constrained multi-objective reinforcement learning (CMORL) as a constrained optimization problem (COP), which theoretically offers strong convergence guarantees. However, the paper lacks detailed insights into the practical implementation of this approach. In contrast, methods like TRPO provide clear derivations of practical algorithms from theoretical foundations, taking into account finite sample sizes and arbitrary parameterizations in deep reinforcement learning. Can you include a similarly detailed derivation to bridge the theoretical framework with practical implementation?

2. **Inclusion of Pareto-Dominated Points in Figure 7**: In Figure 7, numerous points are shown that are Pareto-dominated by others. This raises questions about the relevance of including these dominated points, as they do not contribute to optimal trade-offs and may complicate the interpretation of the results. What is the rationale behind the inclusion of these Pareto-dominated points, and how do they add value to the analysis presented in this figure?

3. **Inconsistencies in Hypervolume (HV) Reporting in Figure 8**: In Section 6.4, Why do the results for Hopper and HalfCheetah in Figure 8 differ significantly from the original results reported in PD-MORL?

4.***Practical Implementation Challenge: High Dimensionality and Computational Expense:***
In the theoretical work, Equation (9) utilizes the inverse Fisher information matrix over the policy parameter theta, potentially resulting in high dimensionality and significant computational expense in practical applications. How does CoMOGA handle this challenge in its practical implementation?

---

> ### Author Response · Authors · 2024-11-22
> **Response by Authors**
>
> We sincerely appreciate the reviewer's insightful comments.
> In the modified paper, we have conducted additional experiments to address common concerns raised by the reviewers.
> Please refer to the general response above for a detailed summary of the newly included experiments.
> The followings are detailed responses to the points raised by Reviewer 6dPa.
>
> **Weakness 1**
> > With PD-MORL reporting HV values in the unit of 1e6 , whereas this paper uses a unit of 0.1 for HV. These inconsistencies create confusion and lead to the perception that PD-MORL performs worse in this comparison than reported in the original study.
>
> We appreciate the reviewer’s comment regarding the difference in hypervolume (HV) units.
> To clarify, as noted in the Metrics part of Section 6.1, we normalized the objectives such that the minimum and maximum values of each objective were scaled to $0$ and $1$, respectively, before computing the hypervolume.
> This normalization was performed to mitigate the impact of varying scales across objectives.
>
> For example, if the scale of the first objective is $100$ times larger than that of the second objective, the hypervolume metric would be dominated by how well the obtained Pareto optimal set covers the first objective, rather than reflecting the performance across all objectives.
> By normalizing the objectives, we resolved this scale issue, resulting in HV values with a maximum theoretical value of $1$, which corresponds to the unit of $0.1$ in our results.
>
> To address the reviewer’s concern and provide additional transparency, we have included HV results computed without normalization in Figure 14 of Appendix D of the modified manuscript.
> We hope this additional data clarifies any potential confusion.
>
> **Weakness 2**
> > HV values are reported across training steps up to 10 e6, while in this study, the results are shown only from 1e6 up to 3e6 steps, Which may not reach the convergency.
>
> We appreciate the reviewer’s comment regarding the training steps for the hypervolume (HV) results and would like to clarify a few points.
>
> First, our experiments were conducted on MO-Gymnasium tasks [1], a well-known MORL benchmark, whereas PD-MORL utilized customized Mujoco tasks that were modified to have multiple objectives.
> This difference in experimental environments may account for the variations in reported HV scales.
>
> Second, in Figure 5 of the MO-Gymnasium paper [1], the authors reported results for the Half-Cheetah task after training MORL algorithms for only 200k training steps.
> Based on this precedent, we believe that the 1M training steps used in our experiments are sufficient for evaluation.
> Additionally, the number of training steps in our study was determined by observing when the performance of our proposed algorithm converged.
>
> Finally, all algorithms in our study were run for the same number of training steps, ensuring fairness in the comparisons.
> We hope this explanation addresses the reviewer’s concerns.
>
> **Question 1**
> > Limited Practical Implementation Details in Constrained Optimization Problem (COP): Can you include a similarly detailed derivation to bridge the theoretical framework with practical implementation?
>
> We appreciate the reviewer’s valuable comment regarding the practical implementation details of solving the constrained optimization problem (COP) in our proposed approach.
> To address this, we have added a detailed explanation of the implementation process for Equations (5) and (6) in Appendix C.3.1 of the revised manuscript.
>
> Briefly, the quadratic programming (QP) problems in Equations (5) and (6) hold strong duality, allowing us to transform them into a lower-dimensional dual space with a dimensionality of $N+M$.
> In this dual space, we solve for the dual solution using a well-established QP solver.
> The obtained dual solution is then mapped back to the primal space to compute the final aggregated gradient.
>
> The appendix provides further details on key implementation steps, such as the computation of the inverse of the Fisher information matrix and the gradients of the objective and constraint functions.
> We hope these additional details address the reviewer’s concerns.
>
> If there are any aspects requiring further elaboration, we would be happy to provide additional clarification.

---

> ### Author Response · Authors · 2024-11-22
> **Response by Authors #2**
>
> **Question 2**
> > Inclusion of Pareto-Dominated Points in Figure 7: What is the rationale behind the inclusion of these Pareto-dominated points, and how do they add value to the analysis presented in this figure?
>
> We appreciate the reviewer’s insightful question regarding the inclusion of Pareto-dominated points in Figure 7 and the potential challenges in interpreting these results.
> We would like to provide clarification on the rationale behind this visualization.
>
> In Figure 7, we visualized the CP front obtained for each algorithm.
> Since the training process was performed with five different random seeds for each algorithm, the CP fronts obtained from each seed (denoted as $P_i$ for the CP front from the $i$-th seed) were visualized together in the same graph.
> As a result, it is possible for points in $P_i$ (the CP front from the $i$-th seed) to be Pareto-dominated by points in $P_j$ (the CP front from the $j$-th seed) even for the same algorithm.
>
> As the reviewer suggested, to report only Pareto-optimal points, one could compute the union of all CP fronts across seeds ($P = \mathbf{PF}(P_1 \cup P_2 \cup \dots \cup P_5)$, where $\mathbf{PF}$ denotes the Pareto front) and visualize this combined Pareto front.
> However, this approach would obscure valuable information about the sensitivity of the algorithm’s performance to random seeds, which is a critical aspect of understanding algorithm robustness.
> Reporting statistical variations, including seed sensitivity, is a standard practice for presenting experimental results.
>
> Nonetheless, we acknowledge that including Pareto-dominated points may complicate the interpretation of results.
> To help interpret Figure 7, we propose the following perspectives:
>
> 1.	Coverage of the Upper-Right Region: The extent to which the CP front covers the upper-right region of the graph indicates how well the algorithm captures Pareto-optimal policies.
> 2.	Self-Dominated Points: If there are numerous self-dominated points (points dominated by other results from the same algorithm), this suggests that the algorithm’s performance is highly sensitive to the choice of random seed.
>
> We believe these perspecitves help to interpret Figure 7 while preserving the valuable information about seed sensitivity.
>
> **Question 3**
> > Inconsistencies in Hypervolume (HV) Reporting in Figure 8: Why do the results for Hopper and HalfCheetah differ significantly from the original results reported in PD-MORL?
>
> We appreciate the reviewer’s question regarding the differences in performance results for Hopper and Half-Cheetah tasks between our study and the results reported in PD-MORL.
> We believe that the main reason for these differences lies in the differences in the training environments used in the two studies.
> As mentioned earlier, we conducted our experiments using MO-Gymnasium [1], a well-established MORL benchmark, whereas PD-MORL used customized Mujoco tasks based on the OpenAI Gym environment.
> Below, we provide a detailed explanation of the differences between the two studies for the Hopper and Half-Cheetah tasks.
>
> 1.	Hopper: In MO-Gymnasium, the Hopper task consists of three objectives: velocity, height, and energy. In contrast, PD-MORL implemented the Hopper task with only two objectives: velocity and energy. This difference in the number of objectives may contributed to the observed performance discrepancies.
> 2.	Half-Cheetah: Both MO-Gymnasium and PD-MORL implement the Half-Cheetah task with two objectives: velocity and energy. However, a key difference lies in the reward formulation for energy: in MO-Gymnasium, the energy reward is negative, while in PD-MORL, it is positive. Negative rewards for energy in MO-Gymnasium can cause the agent to fall into local optima, where it terminates the episode immediately to minimize energy consumption. PD-MORL algorithms may be more sensitive to such reward formulations, leading to a performance decline in this case.
>
> We interpret these differences in reward structures and objective configurations as the primary reasons for the difference in results. We hope this explanation provides clarity regarding the observed discrepancies.

---

> ### Author Response · Authors · 2024-11-22
> **Response by Authors #3**
>
> **Question 4**
> > High Dimensionality and Computational Expense: In the theoretical work, Equation (9) utilizes the inverse Fisher information matrix over the policy parameter theta, potentially resulting in high dimensionality and significant computational expense in practical applications. How does CoMOGA handle this challenge in its practical implementation?
>
> We appreciate the reviewer’s insightful question regarding the computational challenges associated with the inverse Fisher Information Matrix (FIM) in Equation (9).
> As the reviewer correctly pointed out, solving Equation (9) requires computing $H^{-1}g$, where $H$ is the FIM.
> This challenge is not unique to our method and is similarly encountered in TRPO [2].
> Following TRPO’s approach, we adopted the conjugate gradient (CG) method [3] to effectively approximate this computation.
>
> To provide additional insights into the practical implications of this computation, we measured the computation time and GPU memory usage for each module of CoMOGA.
> These results are reported in Appendix D.2.
> As shown, the FIM-related computations account for approximately 10% of the total computation time, indicating that the impact on overall training efficiency is relatively minor.
>
> We hope this clarification and the provided insights address the reviewer’s concern.
>
> **References**
> - [1] Felten, Florian, et al. "A toolkit for reliable benchmarking and research in multi-objective reinforcement learning." Advances in Neural Information Processing Systems 36 (2023).
> - [2] Schulman, John, et al. "Trust region policy optimization." Proceedings of the 32nd International Conference on International Conference on Machine Learning-Volume 37. 2015.
> - [3] Hestenes, Magnus Rudolph, and Eduard Stiefel. Methods of conjugate gradients for solving linear systems. Vol. 49. No. 1. Washington, DC: NBS, 1952.

---

> > ### Comment · Reviewer_6dPa · 2024-11-27
> >
> > Thanks for the detailed response. I have adjusted the score accordingly. Good luck

---

> > > ### Author Response · Authors · 2024-11-28
> > > **Author Response**
> > >
> > > We sincerely appreciate you taking the time to review our response and adjust the score. Thank you for your consideration.

---

### Official Review · Reviewer_em3T · 2024-10-30

**Soundness:** 2
**Presentation:** 3
**Contribution:** 2
**Rating:** 6
**Confidence:** 4

**Summary:**

This paper presents the Constrained Multi-Objective Gradient Aggregator (CoMOGA) algorithm for constrained multi-objective reinforcement learning (CMORL). It transforms the CMORL problem into a constrained optimization framework and addresses the optimization objective within a local region, using constraints designed to enhance the original objectives in proportion to preference values. The authors also provide a convergence guarantee in the tabular setting. Experimental results across several environments demonstrate that CoMOGA achieves constraint satisfaction.

**Strengths:**

## Originality and Significance
1) This paper presents a constrained multi-objective optimization algorithm, CoMOGA, which transforms objectives into constraints. The idea appears to be novel.
2) Multi-objective tasks and constraint optimization are critical issues in reinforcement learning, significantly enhancing the safety of reinforcement learning and its applicability in real-world scenarios.

## Quality and Clarity
1) The paper is well-written and easy to read.
2) This paper includes both theoretical analysis and detailed empirical results.
3) Multiple figures are provided, which are helpful in understanding the main concepts presented in the paper.

**Weaknesses:**

## Framework
1) The core idea of this paper is to avoid gradient conflicts by converting multi-objectives into constraint terms. However, during the optimization process, due to factors such as the linear approximation of the objectives and bias in the value function estimation, it is difficult to avoid constraint violations by the constraint terms of the objective, thus causing gradient conflicts. This may undermine the authors' core contribution.
2) Transforming multiple objectives into constraint terms can complicate the constraint boundaries, thereby increasing the difficulty of finding an optimal policy that satisfies these constraints.
3) It is not clear whether this transformation yields positive gains in computational complexity and convergence rate.

## Experiments
1) There is a lack of sensitivity analysis concerning hyperparameters.
2) The fairness of the ablation experiments is questionable. The constraint optimization algorithm employed for the CAGrad algorithm in the ablation experiments is the Lagrangian method, which is inconsistent with the optimization algorithm used in CoMOGA. The discrepancy in the experimental results may arise from the optimization algorithm itself, thus failing to accurately assess the effectiveness of the core components in CoMOGA.

**Questions:**

1) Can the authors analyze the computational complexity and the rate of convergence of the algorithm after converting multiple objectives into constraints?

2) Does the algorithm converge in continuous space?

---

> ### Author Response · Authors · 2024-11-22
> **Response by Authors**
>
> We sincerely appreciate the reviewer's insightful comments.
> In the modified paper, we have conducted additional experiments to address common concerns raised by the reviewers.
> Please refer to the general response above for a detailed summary of the newly included experiments.
> The followings are detailed responses to the points raised by Reviewer em3T.
>
> **Weakness 1**
> > It is difficult to avoid constraint violations by the constraint terms of the objective, thus causing gradient conflicts.
>
> We appreciate the reviewer’s insightful comment regarding potential constraint violations and gradient conflicts arising from the proposed approach.
> As the reviewer pointed out, the policy update using Equation (5) considers not only the safety constraints but also the objectives.
> Consequently, the updated policy may occasionally violate the safety constraints, in which case we apply the recovery step using Equation (6) to ensure the policy satisfies them.
>
> To clarify, the solution of Equation (5) does not cause gradient conflicts (as the inner product between the solution and the gradient of the objective functions is always non-negative) under the assumption that there is no bias in the value function approximation.
> However, during the recovery step (updating the policy by solving Equation (6)), some objective values can decrease due to the focus only on satisfying the safety constraints.
>
> There are two potential approaches to mitigate this issue.
> The first is to update the policy so that constraint violations never occur.
> However, as the reviewer mentioned, this is challenging in practice due to estimation bias in value function approximation.
> The second approach is to incorporate the objective functions into the gradient aggregation even during the recovery step.
> Nonetheless, as shown in the proof of Theorem 4.2, it is essential to focus exclusively on the safety constraints during the recovery step to ensure convergence to CP-optimal solutions.
> Therefore, incorporating the objective functions during the recovery step is also non-trivial.
> We believe addressing this limitation in future research could lead to an interesting and impactful contribution.
>
> **Weakness 2**
> > Transforming multiple objectives into constraint terms can complicate the constraint boundaries, thereby increasing the difficulty of finding an optimal policy that satisfies these constraints.
>
> We appreciate the reviewer’s comment regarding the potential complication of constraint boundaries when transforming multiple objectives into constraint terms.
> However, we would like to offer a different perspective on this point.
>
> First, it is challenging to determine whether solving an optimization problem with a single objective and $N+M$ constraints is more difficult than solving an optimization problem with $N$ objectives and $M$ constraints.
> Both formulations have their complexities, and the relative difficulty may depend on the specific problem setup.
>
> Second, the policy update using Equation (5), which transforms the objectives into additional constraints, is formulated as a quadratic programming (QP) problem.
> Solving QP problems is well-established and computationally efficient.
> As discussed in Lines 287–294 of the manuscript, this QP can be solved in the dual space, which has a dimensionality of $N+M$.
> Given that this dimensionality is relatively low, the computational difficulty of solving the QP is further reduced.
>
> Based on these considerations, we believe the difficulty introduced by our approach is minimal.
> Nevertheless, we are open to any additional insights or clarifications from the reviewer regarding this concern.
>
> **Weakness 3**
> > There is a lack of sensitivity analysis concerning hyperparameters.
>
> We appreciate the reviewer’s comment regarding the need for a sensitivity analysis of hyperparameters.
> Our proposed algorithm introduces two new hyperparameters: (1) the local region size and (2) the number of preference samples (denoted as $P$ in Algorithm 1).
> To address this concern, we conducted additional experiments on the quadrupedal locomotion task to analyze the impact of these hyperparameters.
> The results have been added to Appendix D.1 of the modified manuscript, and we provide a brief summary below:
>
> 1.	Local Region Size: Increasing the local region size leads to faster convergence. However, larger sizes also cause greater fluctuations in the safety constraints, making it more difficult to train the policy stably.
> 2.	Preference Samples: A higher number of preference samples improves the learning performance. However, as expected, increasing the number of samples results in more iterations, thereby slowing down the training process.
>
> We hope this sensitivity analysis provides further clarity and addresses the reviewer’s concern.
> Please refer to the appendix for a detailed discussion and experimental results.

---

> ### Author Response · Authors · 2024-11-22
> **Response by Authors #2**
>
> **Weakness 4**
> > The fairness of the ablation experiments is questionable.
>
> We appreciate the reviewer’s insightful comment regarding the fairness of the ablation experiments.
> We acknowledge this concern and have revised the ablation study accordingly.
> Before explaining the modified ablation study, we note that “CAGrad+Lagrangian” is now treated as one of the baseline methods and has been included in the main experimental results in the revised manuscript.
>
> The core contributions of our proposed method can be summarized as: (1) transforming objectives into constraints to avoid gradient conflicts, and (2) handling safety constraints by solving a QP problem, where the constraints are linearly approximated within the local region.
> To evaluate the efficiency of these components, we designed two ablation methods:
> 1. CoMOGA+LS: This method excludes (1) and uses linear scalarization to handle multiple objectives.
> 2. CoMOGA+Lag: This method excludes (2) and handles constraints using the Lagrangian method.
>
> The results of this ablation study are included in Section 6.5, and a brief summary is:
> - CoMOGA: Achieves the best performance among the ablation methods in terms of both performance and constraint satisfaction.
> - CoMOGA+Lag: The use of Lagrange multipliers led to instability in handling the safety constraints.
> - CoMOGA+LS: Although the safety constraints were satisfied stably, the Hypervolume was lower than that of CoMOGA, confirming the effectiveness of CoMOGA’s approach to avoiding gradient conflicts.
>
> We hope these additional analyses provide a clearer understanding of the effectiveness of the core components and address the reviewer’s concern thoroughly.
>
> **Question 1**
> > Can the authors analyze the computational complexity and the rate of convergence of the algorithm after converting multiple objectives into constraints?
>
> We appreciate the reviewer’s question regarding the computational complexity and convergence rate of the algorithm after converting multiple objectives into constraints.
> Below, we provide a detailed analysis addressing these aspects.
>
> Our algorithm solves a QP problem in each policy update iteration, where the objectives are transformed into constraints.
> Also, as demonstrated in the proof of Theorem 4.3, the proposed method follows the general policy update rule described in Theorem 4.2.
> To answer the reviewer’s question, we analyzed the computational complexity of the QP problem solved at each update iteration and the convergence rate of the general policy update rule.
>
> First, the computational complexity of solving the QP problem is detailed in Appendix C.3.1.
> Briefly, the main computational overhead arises from transforming the QP problem into the dual space, which scales proportionally to $(N+M)^2$, where $N$ is the number of objectives and $M$ is the number of constraints.
> Second, the convergence rate of the general policy update is discussed in Theorem B.2 in Appendix B.4.
> The theorem establishes that the performance gap between the learned policy and the optimal policy, as well as the constraint violation, are both bounded by $O(1/\sqrt{T})$, where $T$ is the number of policy update iterations.
>
> We hope this explanation and the detailed analyses provided in the appendices address the reviewer’s question comprehensively.
>
> **Question 2**
> > Does the algorithm converge in continuous space?
>
> We appreciate the reviewer’s question regarding the convergence of the algorithm in continuous space.
> Currently, our algorithm does not provide a convergence proof for continuous space.
> To the best of our knowledge, even in the field of Multi-Objective Reinforcement Learning (MORL), algorithms with convergence guarantees in continuous space are rare or nonexistent.
>
> As is common practice in both constrained RL [1, 2] and multi-objective RL [3, 4], convergence is typically proven in tabular settings, and experimental results are used to demonstrate the algorithm’s applicability in continuous space.
> We followed this convention in our work, proving convergence in a tabular setting and validating the algorithm’s performance empirically in continuous space through experiments.
>
> **References**
> - [1] Xu, Tengyu, Yingbin Liang, and Guanghui Lan. "Crpo: A new approach for safe reinforcement learning with convergence guarantee." International Conference on Machine Learning. PMLR, 2021.
> - [2] Bai, Qinbo, et al. "Achieving zero constraint violation for constrained reinforcement learning via primal-dual approach." Proceedings of the AAAI Conference on Artificial Intelligence. Vol. 36. No. 4. 2022.
> - [3] Lu, Haoye, Daniel Herman, and Yaoliang Yu. "Multi-objective reinforcement learning: Convexity, stationarity and pareto optimality." The Eleventh International Conference on Learning Representations. 2023.
> - [4] Basaklar, Toygun, Suat Gumussoy, and Umit Ogras. "PD-MORL: Preference-Driven Multi-Objective Reinforcement Learning Algorithm." The Eleventh International Conference on Learning Representations.

---

> > ### Author Response · Authors · 2024-11-26
> > **Discussion Reminder and Additional Clarifications Regarding Computational Complexity**
> >
> > We would like to remind you that the discussion session will end soon.
> > In addition, we want to provide a more detailed response to the third weakness in the framework raised by Reviewer em3T.
> >
> > > It is not clear whether this transformation yields positive gains in computational complexity and convergence rate.
> >
> > The theoretical analysis related to computational complexity and convergence rate has been addressed in our response to Question 1 above.
> > However, we acknowledge that the analyses do not claim to offer computational speed improvements compared to existing methods.
> > To empirically demonstrate its computational complexity, we compare the training time of our method with other algorithms in Table 2.
> > Furthermore, the computation time of each component in our approach, including the additional time introduced by the transformation, is detailed in Table 6.
> >
> > From Table 6, it can be observed that the additional computation time due to the transformation is relatively minor compared to the overall computation time.
> > Thus, we do not consider the transformation process to be a significant bottleneck in the current experimental settings.
> > However, we agree that if the number of objectives and constraints increases significantly beyond our current setting (less than or equal to five), the transformation could become a bottleneck.
> > This highlights the need for future research to improve computational efficiency for such scenarios.
> >
> > Lastly, we would like to emphasize that the primary purpose of introducing the transformation was not to achieve computational speedups but to ensure stable constraint satisfaction and improved performance.
> > These benefits are demonstrated in the legged robot tasks and in the Safety Gymnasium tasks.
> >
> > We hope this additional explanation clarifies the intention and evaluation of our proposed method.

---

> > > ### Comment · Reviewer_em3T · 2024-12-02
> > >
> > > Thank you for your response!
> > >
> > > I have read your reply and adjusted my score accordingly.

---

> > > > ### Author Response · Authors · 2024-12-02
> > > > **Clarification Regarding Score Adjustment**
> > > >
> > > > Dear Reviewer em3T,
> > > >
> > > > Thank you for taking the time to review our response and provide your feedback.
> > > >
> > > > However, the score does not appear to have been adjusted in your official review.
> > > > This might be an oversight, and we kindly ask if you could verify this at your convenience.
> > > >
> > > > Regardless, we sincerely appreciate your response and engagement with our work.

---

> > > > > ### Author Response · Authors · 2024-12-03
> > > > > **Follow-up by Authors**
> > > > >
> > > > > Dear Reviewer em3T,
> > > > >
> > > > > We noticed that the score does not appear to have been adjusted in your official review.
> > > > > This might be an oversight, and we kindly ask if you could verify this as soon as possible, as the reviewer’s response period is ending soon.

---

> > > > > > ### Comment · Reviewer_em3T · 2024-12-03
> > > > > >
> > > > > > Apologies, I have already updated my official comments.

---

> > > > > > > ### Author Response · Authors · 2024-12-04
> > > > > > > **Author Response**
> > > > > > >
> > > > > > > Thank you for your prompt response and for updating the official comment.
> > > > > > > We appreciate your time and effort in reviewing our manuscript and addressing this matter.

---

> ### Author Response · Authors · 2024-11-30
> **Follow-Up on Author Responses to Reviewer em3T's Comments**
>
> Dear Reviewer em3T,
>
> In response to your valuable feedback, we have provided detailed answers to the weaknesses and questions you highlighted, and we conducted additional experiments to address them thoroughly.
>
> We kindly ask for your feedback on our revisions and responses, as it would mean a lot to us to know whether we have sufficiently addressed your concerns.
> Your input is crucial, and we greatly value your time and effort in helping us improve our work.

---

### Official Review · Reviewer_BUGD · 2024-11-04

**Soundness:** 3
**Presentation:** 3
**Contribution:** 3
**Rating:** 8
**Confidence:** 3

**Summary:**

This paper presents CoMOGA, an algorithm designed to optimize multiple objectives while satisfying constraints. CoMOGA converts objectives into constraints and combines the gradients from both objectives and constraints. Theoretical analysis provided in the paper demonstrates that CoMOGA can achieve a local Pareto optimal policy. Empirical tests show that CoMOGA surpasses competing methods in achieving better coverage of the constrained-Pareto front.

**Strengths:**

- The problem addressed is highly relevant and has practical real-world applications.
- The paper is well-written and easy to understand.
- The experiments are thorough, and the evaluation metrics are clearly defined and articulated.

**Weaknesses:**

- The section on gradient aggregation could be improved by including comparisons with similar MORL methods like CAGrad and PCGrad.

**Questions:**

- Why does CoMOGA perform significantly better than the combination of CAGrad and Lagrangian in Section 6.5? Could the authors expand on this?
- Why is the combination of CAGrad and Lagrangian treated as an ablation study rather than a baseline?
- How are preferences sampled in Line 328, for example, are they uniformly sampled? How does this affect the distribution of results?
- Are the critics trained using the same targets, such as Retrace, across all methods in the experiments?

---

> ### Author Response · Authors · 2024-11-22
> **Response by Authors**
>
> We sincerely appreciate the reviewer's positive and insightful comments.
> In the modified paper, we have conducted additional experiments to address common concerns raised by the reviewers.
> Please refer to the general response above for a detailed summary of the newly included experiments.
> The followings are detailed responses to the points raised by Reviewer BUGD.
>
> **Weakness 1**
> > The section on gradient aggregation could be improved by including comparisons with similar MORL methods like CAGrad and PCGrad.
>
> Thank you for the insightful suggestion.
> In order to reflect the suggestion, we have added a comparison of our proposed algorithm with existing conflict-averse MTL algorithms, such as PCGrad [1] and CAGrad [2], in Appendix E, along with illustrative figures.
> Additionally, we have added a note in the gradient aggregation section of the main text, pointing readers to the appendix for these comparisons.
>
> To summarize the comparison briefly, PCGrad projects each gradient into a region that avoids conflicts before aggregating them, while CAGrad maximizes the minimum improvement across objectives.
> In contrast, our proposed method aims to maximize all objectives uniformly.
> Also, the proposed method can seamlessly integrate safety constraints, offering a distinct advantage in the constrained multi-objective RL field.
>
> **Question 1**
> > Why does CoMOGA perform significantly better than the combination of CAGrad and Lagrangian in Section 6.5? Could the authors expand on this?
>
> While both methods adopt conflict-averse approaches, the key difference lies in how the safety constraints are handled.
> The proposed CoMOGA method addresses the safety constraints by solving a quadratic programming (QP) problem within a local region, which eliminates the need for additional optimization variables.
>
> In contrast, the "CAGrad + Lagrangian" method involves simultaneously learning both the policy and the Lagrange multipliers.
> This dual optimization can lead to instability during training, as updates to the policy and multipliers may interfere with each other.
> This instability becomes particularly critical in environments with numerous constraints, such as legged robot locomotion tasks, due to the large number of Lagrange multipliers.
> We believe this difference in constraint handling is the primary reason CoMOGA outperforms "CAGrad + Lagrangian".
>
> Additionally, we have revised the manuscript to include an ablation study comparing CoMOGA with CoMOGA + Lagrangian ("CoMOGA + Lag") and CoMOGA with linear scalarization ("CoMOGA + LS").
> The results show that "CoMOGA + Lag" is less stable during training than "CoMOGA" but still outperforms "CAGrad + Lagrangian".
> This is likely because "CoMOGA + Lag" update the policy within a local region, which helps mitigate the instability caused by the multipliers.
>
> **Question 2**
> > Why is the combination of CAGrad and Lagrangian treated as an ablation study rather than a baseline?
>
> Thank you for pointing out this issue.
> We have included "CAGrad + Lagrangian" as a baseline algorithm instead of treating it as part of the ablation study.
> The results of training with "CAGrad + Lagrangian" as a baseline can be found in Section 6 of the revised manuscript.
>
> For the ablation study, we have revised the study to analyze the effects of the core components of our proposed method, as suggested by Reviewer em3T.
> Details regarding this analysis can be found in the general response.

---

> ### Author Response · Authors · 2024-11-22
> **Response by Authors #2**
>
> **Question 3**
> > How are preferences sampled in Line 328, for example, are they uniformly sampled? How does this affect the distribution of results?
>
> In our implementation, preferences are sampled using a hybrid approach: with a 50% probability, preferences are sampled as one-hot vectors (where the dimension with a value of one is chosen uniformly at random), and with the remaining 50%, they are sampled from a Dirichlet distribution.
> This choice was made because, while the standard practice in MORL field is to sample preferences from a Dirichlet distribution, doing so reduces the likelihood of sampling cases where a single objective is strongly preferred.
> To ensure that such cases are adequately considered, we adopted this hybrid sampling strategy.
>
> To address the reviewer’s question about the impact of the sampling method on the results, we conducted additional experiments on the legged locomotion tasks, comparing three sampling strategies: (1) sampling only one-hot vectors, (2) sampling only from the Dirichlet distribution, and (3) our hybrid approach of combining one-hot and Dirichlet sampling.
> The results of these experiments are reported in Appendix D.3 of the modified manuscript.
> In summary, sampling from the Dirichlet distribution yields the best overall performance.
> The hybrid approach performs nearly as well as Dirichlet sampling, while the one-hot sampling method results in significant performance degradation.
> These findings suggest that the Dirichlet distribution, which enables uniform sampling across the preference space, is the most effective approach.
>
> **Question 4**
> > Are the critics trained using the same targets, such as Retrace, across all methods in the experiments?
>
> Since LP3 and our proposed method update the critic at the end of each episode, we utilized the Retrace algorithm for calculating the critic’s targets.
> However, PD-MORL and CAPQL are based on TD3 and SAC, respectively, which update the critic at every step.
> Due to this stepwise update mechanism, using Retrace, which requires episode-wise computation of the critic’s targets, is impractical for these methods.
> Instead, we employed one-step TD targets for PD-MORL and CAPQL.
> This is described in detail in Appendix C.3.3.
>
> **References**
> - [1] Yu, Tianhe, et al. "Gradient surgery for multi-task learning." Advances in Neural Information Processing Systems 33 (2020): 5824-5836.
> - [2] Liu, Bo, et al. "Conflict-averse gradient descent for multi-task learning." Advances in Neural Information Processing Systems 34 (2021): 18878-18890.

---

> > ### Comment · Reviewer_BUGD · 2024-11-27
> > **Re**
> >
> > Thank the authors for the detailed reply. I will maintain my current rating of this paper and I am open to further discussion with other reviewers.

---

> > > ### Author Response · Authors · 2024-11-28
> > > **Author Response**
> > >
> > > We sincerely appreciate you taking the time to review our response and maintaining such a positive stance.

---

### Official Review · Reviewer_w9r3 · 2024-11-04

**Soundness:** 3
**Presentation:** 3
**Contribution:** 3
**Rating:** 6
**Confidence:** 3

**Summary:**

The paper presents an algorithm for multi objective reinforcement learning under constraints. The authors first identified that
maximizing a specific return within the trust region is equivalent to minimizing $||\Delta \theta||^2_H$$ under a constraint that the
return increases (if the solution exists). Therefore, the multi objective RL under constraints can be formulated as a problem that
minimizes the induced norm of $\Delta \theta$ under a bunch of constraints that correspond to the multiple task returns and
constraints. By using a first-order approximation of the constraints, the problem reduces to a quadratic programming problem. To avoid
the violation of the constraint due to the inaccurate linear approximation, the authors propose to remove the return constraints once
the violation happens.

The above can be generalized to multiple objectives with preferences, and then to learn a universal policy (i.e., a policy that
is conditioned on any given preference $w \in \R^N$), the authors propose to learn a policy that minimize the expected KL divergence
between the learned policy conditioned on a specific preference and the updated policy that solves the above quadratic programming. In
other word, it amortizes the optimization into a learned conditional policy.

The authors conduct experiments on both synthetic toy and constrained RL problems in simulation, demonstrating that their method
results in superior performance (in terms of optimizing the objective and satisfying constraints) compared against prior methods. The
authors also provide a convergence guarantee for the proposed method.

**Strengths:**

1. The problem of interest has broad application.

2. The proposed method is straightforward yet well motivated. Specifically, the transformation of objective to constraint is
convenient.

3. The presentation is clear, and the figures illustrate the main ideas well.

4. The empirical results demonstrate that the proposed method consistently outperforms existing methods.

**Weaknesses:**

One main weakness is the computational efficiency of the proposed method, which might hinder the practical usage of the proposed
method. First it requires computing/storing gradients to constraints/rewards, which can be quite large if N and M are large. Second, it
requires estimating terms like $x^\top H x$ where x is the fisher information matrix, while this does not in general require computing
$H$ explicitly, it sill requires computing the inner product like $x^\top \nabla \log \pi$ (which might need extra space). The authors
did not elaborate on this aspect, and there is no time complexity evaluation in the experiments or analysis sections.

**Questions:**

1. I think there is a gap from (3) to (4) where the authors generalize to multiple objectives. In the multiobjective case (without
constraints), are (2) and (3) equivalent only when a solution exists?  If so, is it the case that the norm constraint can always
be satisfied, while the monotonic improvement constraints are not?

2. The scaling of the gradient essentially adds the norm constraint back, but if this is the case, why not start with that
constraint and use the form (2)? Like max_d d^\top g_i, s.t., b^\top d + J < d_k, ||d||_H \leq \epsilon? Is it because this form is not
easily solvable due to the norm constraint (difficult to estimate H^{1/2})? If so, the authors could explicitly mention this.

3. The authors claim that using (6) can help correct the policy when it avoids any constraint due to solving (5) which uses linear
approximation. But isn't (6) also using linear approximation, and the solution to (5) should also satisfy constraints in (6) if there
exists any feasible solution? Can the authors elaborate on what this means exactly?

4. The first sentence of the abstract asserts that an RL agent needs to consider multiple objectives.  That's not necessarily the
case - the reward hypothesis states that an agent's objective can always be summarized with a single scalar reward function.  How
would you motivate this paper to someone who fundamentally disagrees that an agent needs to consider multiple objectives separately
(as opposed to aggregating them to explicitly encode their tradeoffs)?

---

> ### Author Response · Authors · 2024-11-22
> **Response by Authors**
>
> We sincerely appreciate the reviewer's positive and insightful comments.
> In the modified paper, we have conducted additional experiments to address common concerns raised by the reviewers.
> Please refer to the general response above for a detailed summary of the newly included experiments.
> The followings are detailed responses to the points raised by Reviewer w9r3.
>
> **Weakness 1**
> > One main weakness is the computational efficiency of the proposed method, which might hinder the practical usage of the proposed method. First it requires computing/storing gradients to constraints/rewards, which can be quite large if N and M are large.
>
> We appreciate the reviewer’s observation regarding the computational efficiency of our proposed method, particularly the need to compute and store gradients for constraint and objective functions, which can become significant when $N+M$ are large.
> As the reviewer noted, this is a common limitation shared by many conflict-averse approaches [1, 2, 3].
>
> To provide further insight into the practical impact of this issue, we analyzed the proportion of training time spent on gradient computation and the memory usage dedicated to gradient storage across different tasks.
> These results are detailed in Appendix D.2 of the revised paper.
> Our analysis indicates that when $N+M \leq 5$, the computational and memory overhead remains relatively small, suggesting that the method is still practical despite a slight reduction in computation speed.
> However, for scenarios where $N+M$ is mutch larger than our settings, the computational cost may become significant.
> In such cases, adopting techniques like stochastic selection of objectives and constraints for gradient computation, as done in several MTL algorithms [2], could be a promising direction to improve computational efficiency.
>
> **Weakness 2**
> > it requires estimating terms like $x^T H x$, where $H$ is the fisher information matrix, while this does not in general require computing $H$ explicitly, it sill requires computing the inner product like $x^T \nabla \log \pi $ (which might need extra space). The authors did not elaborate on this aspect, and there is no time complexity evaluation in the experiments or analysis sections.
>
> As mentioned by the reviewer, computing the Fisher information matrix can be computationally intensive.
> Many trust-region-based methods [4, 5], including our approach, employ approximate techniques such as conjugate gradient methods to estimate terms like $H^{-1}x$.
> Despite these approximations, the issue of computational overhead persists.
>
> To elaborate on this aspect, we have conducted additional analysis to evaluate the proportion of total training time spent on computing the matrix $H$.
> The results of this analysis are added to Appendix D.2 of the revised manuscript.
> Our findings indicate that, under our current settings, the computational cost associated with this step does not constitute a significant bottleneck.
>
> Nevertheless, we acknowledge that in larger or more complex setups, the overhead could increase.
> Exploring alternative approaches, such as approximation methods that do not explicitly compute the Fisher information matrix (e.g., as done in PPO [6]), would be a valuable direction for future work.
>
> **Question 1**
> > I think there is a gap from (3) to (4) where the authors generalize to multiple objectives. In the multi-objective case (without constraints), are (2) and (3) equivalent only when a solution exists? If so, is it the case that the norm constraint can always be satisfied, while the monotonic improvement constraints are not?
>
> The reviewer raises a question about the relationship between Equations (2) and (3).
> As mentioned in the text immediately following Equation (2), the solutions to these two formulations are equivalent under the assumption that the objective functions are linear.
> Without this linearity assumption, the two problems are not equivalent, and as the reviewer pointed out, there could be scenarios where the monotonic improvement constraints are not satisfied while the norm constraint is.
>
> For clarification, the purpose of Section 4.1 was to describe how Equation (5) was derived.
> We acknowledge, as the reviewer has pointed out, that the transition from (2) → (3) → (4) → (5) is not mathematically rigorous or equivalent.
> However, as demonstrated in Section 4.4, even with this heuristic formulation of Equation (5), the policy update can still converge to a CP optimal policy, which supports the practical validity of this approach.

---

> ### Author Response · Authors · 2024-11-22
> **Response by Authors #2**
>
> **Question 2**
> > The scaling of the gradient essentially adds the norm constraint back, but if this is the case, why not start with that constraint and use the form (2)? Like $\max_d d^T g_i$ s.t. $b^T d + J < d_k, ||d||_H \leq \epsilon$? Is it because this form is not easily solvable due to the norm constraint (difficult to estimate $H^{1/2}$)? If so, the authors could explicitly mention this.
>
> We appreciate the reviewer’s insightful comment.
> As the reviewer noted, it is indeed possible to consider starting from Equation (2) by adding the norm constraint.
> However, in multi-objective settings, the most challenging aspect lies in effectively addressing multiple objectives simultaneously.
>
> If we were to start with Equation (2) as suggested, the problem, despite being linear in form, would still involve multiple objectives, making the determination of an optimal gradient direction nontrivial.
> To address this challenge, we adopted an approach that first reformulates the original objective functions as constraints, effectively transforming the multi-objective problem into a single-objective one.
> This leads to Equation (5), which becomes solvable using quadratic programming (QP) solvers.
>
> That said, as long as the obtained policy gradient satisfies Theorem 4.2, it guarantees convergence to a Pareto optimal solution.
> This implies that alternative methods for obtaining the gradient direction could also be explored.
> Identifying more efficient approaches to compute the gradient direction could be an interesting direction for future research.
>
> **Question 3**
> > The authors claim that using (6) can help correct the policy when it avoids any constraint due to solving (5) which uses linear approximation. But isn't (6) also using linear approximation, and the solution to (5) should also satisfy constraints in (6) if there exists any feasible solution? Can the authors elaborate on what this means exactly?
>
> We appreciate the reviewer’s insightful question.
> Strictly speaking, Equation (6) also relies on linear approximation.
> However, as shown in the proof of Theorem 4.2, updating the policy only considering the safety constraints (as in Equation (6)) ensures that the policy can recover to satisfy the safety constraints within a finite number of update iterations.
>
> Additionally, as the reviewer pointed out, the solution to Equation (5) inherently satisfies the constraints in Equation (6) if a feasible solution exists.
> However, the gradients used for policy updates are clipped to remain within the local region.
> As a result, it is more meaningful to focus on the direction of the gradients rather than the obtained gradients themselves.
> Since the solution derived from Equation (5) considers both the safety constraints and the objectives, it is more likely to produce a direction that violates the safety constraints.
> To address this, we performed updates using Equation (6), which focuses solely on satisfying the safety constraints.
>
> To clarify our reasoning for introducing Equation (6), we have revised the manuscript as follows:
> “When both the safety constraints and the objectives are reflected in the updates, the updated policy can violate the safety constraints by overly focusing on objective improvements. To address this issue, when the current policy violates constraints, we update the policy by considering only the safety constraints.”

---

> ### Author Response · Authors · 2024-11-22
> **Response by Authors #3**
>
> **Question 4**
> > The first sentence of the abstract asserts that an RL agent needs to consider multiple objectives. That's not necessarily the case - the reward hypothesis states that an agent's objective can always be summarized with a single scalar reward function. How would you motivate this paper to someone who fundamentally disagrees that an agent needs to consider multiple objectives separately (as opposed to aggregating them to explicitly encode their tradeoffs)?
>
> As noted by the reviewer, RL problems can be formulated using a single reward function, which may lead some to question the utility of a multi-objective RL framework.
> However, we posit that even those who adopt the single reward function approach implicitly consider multiple objectives during the reward design process.
> The key distinction lies in whether these multiple objectives are addressed explicitly or implicitly.
> From this perspective, the first sentence of our abstract remains valid, as it highlights the inherent need to address multiple objectives in some form.
>
> To motivate the explicit treatment of multiple objectives, we have briefly outlined the motivation in the introduction (lines 33–37).
> For more details, designing a reward function to encode multiple objectives often involves iterative process of reward shaping.
> This process typically requires repeatedly modifying the reward function based on the trained behaviors and retraining policies, which can be time-consuming and labor-intensive.
> In contrast, by explicitly addressing multiple objectives, as in our proposed approach, one can learn a policy set that directly encompasses diverse trade-offs.
> This enables practitioners to select the desired policy from the set without additional training, thereby significantly reducing the effort required for reward shaping.
>
> **References**
> - [1] Désidéri, Jean-Antoine. "Multiple-gradient descent algorithm (MGDA) for multiobjective optimization." Comptes Rendus Mathematique 350.5-6 (2012): 313-318.
> - [2] Liu, Bo, et al. "Conflict-averse gradient descent for multi-task learning." Advances in Neural Information Processing Systems 34 (2021): 18878-18890.
> - [3] Yu, Tianhe, et al. "Gradient surgery for multi-task learning." Advances in Neural Information Processing Systems 33 (2020): 5824-5836.
> - [4] Schulman, John, et al. "Trust region policy optimization." Proceedings of International Conference on Machine Learning. 2015.
> - [5] Achiam, Joshua, et al. "Constrained policy optimization." International conference on machine learning. PMLR, 2017.
> - [6] Schulman, John, et al. "Proximal policy optimization algorithms." arXiv preprint arXiv:1707.06347 (2017).

---

> > ### Comment · Reviewer_w9r3 · 2024-11-24
> >
> > Thanks for your responses.  I see the computation comparison in Appendix D.2.  Overall, I continue to rate the paper positively and maintain my score.

---

> > > ### Author Response · Authors · 2024-11-28
> > > **Author Response**
> > >
> > > Thank you for reading our response and for maintaining a positive stance.
> > > We truly appreciate it.

---

### Author Response · Authors · 2024-11-22
**General Response by Authors**

## **General Response**

We appreciate all the reviewers for their insightful comments and suggestions.
In response to the feedback, we conducted various additional experiments and analyses, and revised the manuscript accordingly.
The updated parts in the paper are highlighted in magenta for clarity.
Below is a summary of the modifications made:

1. In the main text, we have added CAGrad+Lagrangian as a baseline, which was previously analyzed as part of the ablation study. In the revised ablation study section, we analyze the core components of CoMOGA by introducing two comparative methods: (1) CoMOGA+LS, which replaces transforming objectives into constraints with linear scalarization (LS), and (2) CoMOGA+Lag, which replaces solving QP problems for constraint handling with the Lagrangian method.

2. A comparison with existing conflict-averse MTL algorithms has been described in Appendix E.

3. Sensitivity analysis for hyperparameters has been included in Appendix D.1.

4. An analysis of the computational time and GPU memory usage of CoMOGA’s components has been added to Appendix D.2.

5. Detailed implementation methods for solving the designed QP problem in Equations 5 and 6 are provided in Appendix C.3.1.

6. A description and analysis of the preference sampling approach in Algorithm 1 have been added to Appendix D.3.

---

### Meta-Review · Area_Chair_yx5d · 2024-12-18

**Metareview:**

This paper studies multi-objective reinforcement learning under constraints, by transforming the problem into a constrained optimization framework and addressing the optimization objective within a local region, with constraints designed to enhance the original objectives in proportion to preference values. Both theoretical and experimental results were provided. The problem studied is well-motivated and significant, the algorithmic idea is novel, and the paper is very well-written and easy to follow. There were some concerns regarding the high computational complexity and implementability of the proposed approaches, and the sufficiency of some experimental results. Overall the merits of the paper outweigh the drawbacks, and I suggest the authors incorporate the feedback from this round in preparing the camera-ready version of the paper.

**Additional Comments On Reviewer Discussion:**

There were some common concerns regarding the sufficiency of the experiments, especially in terms of the comparison with baselines, and the clarity of the computational complexity and certain implementation details. The authors acknowledged the comments, and revised the paper accordingly, which addressed most of the major common comments.

---

### Decision · Program_Chairs · 2025-01-22

Accept (Poster)